



# Particulate organic matter in the Lena River and its Delta: From the permafrost catchment to the Arctic Ocean

Olga Ogneva[1,2,3], Gesine Mollenhauer[1,3], Bennet Juhls[2], Tina Sanders[4], Juri Palmtag[5,a], Matthias Fuchs[2], Hendrik Grotheer[1], Paul J. Mann[5] and Jens Strauss[2]

[1]Marine Geochemistry Section, Alfred Wegener Institute Helmholtz Centre for Polar and Marine Research, Bremerhaven, 27570, Germany
[2]Permafrost Research Section, Alfred Wegener Institute Helmholtz Centre for Polar and Marine Research, Potsdam, 14473, Germany
[3]Faculty of Geosciences, University of Bremen, Bremen, 28359, Germany
[4]Institute for Carbon Cycles, Helmholtz Centre Hereon, Geesthacht, 21502, Germany
[5]Department of Geography and Environmental Sciences, Northumbria University, Newcastle-upon-Tyne, NE1 8ST, UK
[a]now at: Department of Human Geography, Stockholm University, Stockholm, Sweden

*Correspondence to*: Olga Ogneva (Olga.Ogneva@awi.de), Gesine Mollenhauer (Gesine.Mollenhauer@awi.de) and Jens Strauss (Jens.Strauss@awi.de)

**Abstract.** Rapid Arctic warming accelerates permafrost thaw, causing an additional release of terrestrial organic matter (OM) into rivers, and ultimately, after transport via deltas and estuaries, to the Arctic Ocean nearshore. The majority of our understanding of nearshore OM dynamics and fate has been developed from freshwater rivers, despite the likely impact of highly dynamic estuarine and deltaic environments on transformation, storage, and age of OM delivered to coastal waters. Here, we studied OM dynamics within the Lena River main stem and Lena Delta along an approximately ~1600 km long
transect from Yakutsk, downstream to the delta disembogue into the Laptev Sea. We measured particulate organic carbon (POC), total suspended matter (TSM), and carbon isotopes ($\delta^{13}C$ and $\Delta^{14}C$) in POC to compare riverine and deltaic OM composition and changes in OM source and fate during transport offshore. We found that TSM and POC concentrations decreased by 55 and 70 %, respectively, during transit from the main stem to the delta and Arctic Ocean. We found deltaic POC to be strongly depleted in $^{13}C$ relative to fluvial POC, indicating a significant phytoplankton contribution to deltaic POC
(~68 ±6 %). Dual-carbon ($\Delta^{14}C$ and $\delta^{13}C$) isotope mixing model analyses suggested an additional input of permafrost-derived OM into deltaic waters (~18 ±4 % of deltaic POC originates from Pleistocene deposits vs ~ 5 ±4% in the river main stem). Despite the lower concentration of POC in the delta than in the main stem (0.41 ±0.10 vs. 0.79 ±0.30 mg L$^{-1}$, respectively ), the amount of POC derived from Pleistocene deposits in deltaic waters was almost twice as large as POC of Yedoma origin in the main stem (0.07 ±0.02 and 0.04 ±0.02 mg L$^{-1}$, respectively). We assert that estuarine and deltaic processes require
consideration in order to correctly understand OM dynamics throughout Arctic nearshore coastal zones and how these processes may evolve under future climate-driven change.



# 1 Introduction

Arctic rivers contribute substantial quantities of freshwater and organic matter (OM) to the Arctic Ocean, thereby influencing coastal patterns of stratification, ocean chemistry, and biogeochemistry. These freshwater and OM loads originate from an extensive accumulated pan-Arctic watershed area larger than the Arctic Ocean itself (Terhaar et al., 2021). Arctic rivers, therefore, provide an integrated signature of processes and changes occurring across Arctic watersheds. In these Arctic watersheds, soils and permafrost contain between 1460 - 1600 Pg of organic carbon (C: within the upper 25 m depth) (Hugelius et al., 2014; Strauss et al., 2021a), corresponding to about 2.5 times as much C as in the current atmosphere and more than half of the organic C stored in soils globally (Köchy et al., 2015). Annual Arctic air temperatures have risen by more than four times the magnitude of the global mean air temperature rise (Rantanen et al., 2022), with warming in winter being four times greater than in summer (Ballinger et al., 2020). Precipitation is increasing as well and is projected to be >50 % higher by 2100 (Overland et al., 2014). In response to warming air temperatures permafrost is thawing (Biskaborn et al., 2019; Fox-Kemper et al., 2021).

Permafrost can modify fluvial processes and their functioning, for example via the occurrence of surface runoff from hillslopes and sediment erosion in river valleys and channels (Tananaev & Lotsary, 2022). Permafrost thaw has consequences for river discharge in several ways. Deepening of the seasonal active layer and release of waters from melting ground ice result in intensified summer river runoff and increased groundwater storage (Walvoord & Striegl, 2007). In addition, the delayed active layer freeze-up increases winter river runoff (Walvoord & Kurylyk, 2016; Lamontagne-Hallé et al., 2018; Wang et al., 2021a) and enhances riverbank and coastal erosion (Fuchs et al., 2020). Enhanced terrestrial permafrost thaw and intensification of hydrological cycles have the potential to enrich Arctic rivers with remobilised OM and nutrients, modifying food web dynamics and changing the connectivity between terrestrial landscapes and nearshore ecosystems (Brown et al., 2019; Terhaar et al., 2021; Mann et al., 2022). Thus, together with rising temperatures, precipitation, and changes in evapotranspiration, permafrost degradation alters the biogeochemical cycle of the rivers and the freshwater cycle of the Arctic and ultimately modifies river discharge (Carmack et al., 2016; Lique et al., 2016; Vihma et al., 2016; Oliva & Fritz, 2018; Brown et al., 2019; Yang et al., 2021). Permafrost degradation and associated active layer thickening accelerates riverine carbonate, nitrogen, and phosphorus exports (Zhang et al., 2021) and provides additional C of permafrost origin, especially in summer, fall, and winter (Wild et al., 2019). The permafrost mobilised from catchments that enters a river can subsequently undergo a variety of processes. Once mobilised, OM from permafrost is susceptible to transformation and modification during transport (Vonk et al., 2019). Upon discharge into and offshore transport with the Arctic Ocean (Bröder et al., 2018) it is re-buried in marine sedimentary depo-centres, where it is either being removed from the active C cycle or re-mineralised further (Grotheer et al., 2020).

The nearshore coastal zone of the Arctic Ocean (including deltas, estuaries, and coasts) is of great importance as major transformation processes of terrestrial material are expected to take place in these biogeochemically active areas (Tanski et al., 2019; Jong et al., 2020; Sanders et al., 2022). Despite the importance of Arctic estuarine and deltaic environments in OM





biogeochemistry, their functioning is still poorly understood; coastal ocean dynamics are inferred from freshwater endmembers based purely on riverine OM properties. Since 2003, the hydrology and biogeochemistry of the greatest (largest) Arctic rivers (Ob, Yenisei, Lena, Kolyma, Yukon, and Mackenzie) have been measured as part of the Arctic Great Rivers Observatory (ArcticGRO; https://arcticgreatrivers.org/). Historical records together with ArcticGRO data have demonstrated that the long-term increasing freshwater discharge trend has been most pronounced for rivers across the Eurasian Arctic, constituting the

strongest evidence of Arctic freshwater cycle intensification (Feng et al., 2021; Shiklomanov et al., 2021).

Within the framework of the ArcticGRO approximately 5-6 samples per year are collected. Samples characterising the discharge from the largest Arctic rivers are taken directly from the rivers' main stems rather than from their deltas and estuaries; thus, for example, sampling from the Lena River took place at the town of Zhigansk, located ~800 km upstream from the Lena Delta. The long distance of this sampling site from the areas, where the river enters the Arctic Ocean, and the deficit of

information about the delta and the potential biogeochemial processes taking place there (OM transformation/sedimentation/enrichment) may lead to a distortion or a lack of information about the final state of OM reaching the Arctic Ocean.

In this study, we aim to characterise particulate organic C (POC) along the Lena River over a transect from upper reaches of the Lena River near Yakutsk (approximately 1640 km from the coast) north to the Lena Delta in order to decipher the

distribution, main sources, and transformation of particulate organic matter (POM) on its way from the permafrost catchment to the Arctic Ocean. Our findings show that the concentration and composition of the POC pool are highly dynamic during transport and that the transformation and storage of riverine OM need to be accounted for when examining contemporary and projecting future changes in coastal processes.

## 2 Materials and methods

### 2.1 Study area

The Lena River is one of the greatest Arctic rivers. It discharges approximately 543 km$^3$ yr$^{-1}$ of water into the Arctic Ocean (mean annual discharge in the period 1936 – 2019) (Wang et al., 2021b) which is the second-largest amount of water discharged into the Arctic Ocean of all Arctic rivers (Gordeev, 2006; Holmes et al., 2018), and it transports the largest amount of POC (814 ±52x10$^9$ g) of all Arctic rivers (McClelland et al., 2016) into the Laptev Sea. This marginal sea of the Arctic Ocean is a

key region controlling the sea-ice formation and drift patterns (Krumpen et al., 2019). Together with the East Siberian Sea and the Russian part of the Chukchi Sea it constitutes the largest shelf system in the Arctic: The East Siberian Arctic Shelf.

The Lena River is 4400 km long from its origin at 53°N, north of Lake Baikal, to 71°N, where it reaches the Laptev Sea. It drains an area of ∼2.61×10$^6$ km$^2$, of which more than 94 % is assumed to be underlain permafrost (mainly continuous: 70.5 %) (Obu et al., 2019; Juhls et al., 2020). Running from the south to the north of East Siberia, the Lena River receives OM from

various sources within its basin, which is covered mostly by taiga forest (72 % coverage) and tundra ecosystems (12 %) (Amon et al., 2012) and includes Holocene and Pleistocene deposits (Yedoma), which are widespread across the region and cover



approximately 3.5 % of the Lena watershed area (Strauss et al., 2013, 2021b). The delta of the Lena River occupies an area of $28.5 \times 10^3$ km²; it is the largest delta in the Arctic on the Eurasian continent, and one of the biggest deltas in the world (Semiletov et al., 2011). The Lena main stem reaches Stolb Island at the apex of the Lena Delta (Figure 1a); there it divides

into numerous branches and more than 800 transverse channels with a total length of 6500 km, forming the delta (Fedorova et al., 2015). About 80-90 % of Lena River derived water and 85 % of sediments enter the eastern Laptev Sea along two major branches: the Sardakhskaya-Trofimovskaya system (accounting for 60-75 % of Lena River water and up to 70 % of sediment discharge, with the Sardakhskaya branch itself transporting 23-33 %) and the Bykovskaya branch (20-25 % of water and up to 15 % of sediment discharge) (Ivanov & Piskun, 1999; Charkin et al., 2011). The other two main branches are the

Olenekskaya and Tumanskaya (together transporting 5-10 % of water and 10 % of sediment discharge) which flow towards the western and central Laptev Sea.



**Figure 1.** The Lena catchment and its Delta. a) Sampling locations along the Lena River main stem and the Lena River catchment area. Yedoma distribution map is modified after Strauss et al. (2021b); b) Sampling locations along the Sardakhskaya branch in the Lena Delta.

Nowadays, constant patterns of shifting temperature and precipitation regimes are observed in the Lena watershed (Pohl et al., 2020) which is characterised by permafrost and is particularly sensitive to climate change. Thus, over the past 84 years, the discharge of the Lena River has increased by 22 % (Wang et al., 2021b) and reached as much as 626 km$^3$ yr$^{-1}$ for the years 2016-2018 compared to an average of 557 km$^3$ yr$^{-1}$ in the years 1981-2010 (Holmes et al., 2019). Nevertheless, as mentioned above, characteristics of OM discharged from the Lena River are measured by ArcticGRO, which collects water samples near Zhigansk, located ~800 km upstream from the Lena Delta. As this location is far from the site where Lena runoff enters the Arctic Ocean and, in particular, is far south of the delta, any biogeochemical processes taking place in the delta are not reflected in the ArcticGRO data, and the properties of water and suspended materials sampled at Zhigansk may in fact not be entirely representative of the discharge to the ocean.

### 2.2 Sampling

In this study, we sampled along the Lena main stem from Yakutsk to Stolb Island and then onward through the Sardakhskaya branch. Samples from the Lena main stem and its Delta were taken separately during two independent expeditions in 2019. The first sampling campaign took place between 20 June and 30 June 2019 and covered a transect along the Lena main stem. Surface water was taken along a 1300 km long transect from Yakutsk to the Lena Delta (Figure 1a) onboard the vessel Merzlotoved. Sampling took place at nine sites approximately every 150 km on the way to the Lena Delta. This transect intersects the location of the perennial ArcticGRO observation program and includes a sampling station downstream of the town of Zhigansk (station WL19-05). Water samples were taken from the surface (~1 m water depth) using a pre-cleaned plastic bucket. The water sample was stored in 500 ml high-density polyethylene (HDPE) bottles (pre-cleaned with 10 % HCl for 3 days) and immediately frozen at -10 °C. The second sampling campaign started on 7August 2019. The aim of the expedition was a detailed investigation along the Sardakhskaya branch, from Stolb Island to the eastern Laptev Sea (Fuchs et al., 2021). The distance between the sampling locations varied between ~20 km and ~5 km, with coarser sampling at the beginning of the transect near Stolb and increasing spatial resolution towards the Laptev Sea (Figure 1b).

We used a 5 litre water sampler (UWITEC, Austria) to collect water from one to three depths per station (depending on the bottom depth at the sampling location). We took surface water (0-1 m) at each sampling site (or the middle of the water profile is the depth did not exceed 3 m). If the river depth at a location was 3-7 m, two depths were sampled: surface water and above the bottom depth (0-1 m and 3-7 m). When water depth exceeded 7 m, water was sampled at three depths: 1) surface (0-1 m), 2) mid-depth in the water column (3-7 m), and 3) just above the river bottom (8-18 m). The water samples were collected in 20 L plastic canisters, which were pre-cleaned with 10 % HCL and kept cool until return to the laboratory on Samoylov Island, where the samples were further analysed.





Additionally, samples from previous Lena Delta expeditions were used for this study. Those samples were taken near Stolb Island in 2016 (8 August and 15 August) and in 2017 (11 July and 25 July). This collection of samples and further analyses were conducted the same way as for the samples from the deltaic transect in 2019.

### 2.3 Laboratory analyses

**2.3.1 Total suspended matter concentration**

Directly after sampling (delta transect) or after 40 days of freezing (Yakutsk-delta transect), samples were delivered to the Samoylov Island research station laboratory, where they were processed for further analyses. Water was filtered through pre-combusted (4.5 hours, 450 °C) and pre-weighed glass fibre filters (GF/F Whatman, 0.75 µm membrane, Ø 2.5 cm) for total suspended matter (TSM) content, POC concentration, and C isotope analysis. Filters were stored frozen in pre-combusted

glass petri dishes. After filtration the filters were dried for 24 hours at 40 °C and weighed. We used the difference in weights between dried filters with TSM and pre-weighed empty filters to calculate TSM concentration per unit volume of water (mg $L^{-1}$).

**2.3.2 Particulate organic carbon concentration, carbon isotope analyses ($\Delta^{14}C$, $\delta^{13}C$), and relative organic carbon content in total suspended matter**

After calculation of TSM, selected filters (different replicates from the same water sample) were further processed 1) to determine total POC concentration (mg $L^{-1}$) together with stable C isotopes, and 2) to determine $\Delta^{14}C$ of POC. For this purpose, in both cases filters were acidified with 10 % HCl to remove inorganic C (sufficient HCL to wet the filter surface including its sediment). Then they were dried again for 24 hours at 40 °C and packed/rolled into small tin boats. If the amount of C exceeded 100 µg, only a subsample of the filter was used.

POC content on the filter and its $\delta^{13}C$ signature were measured on a Sercon - 20-20 isotope ratio mass spectrometer (IRMS) coupled to an Automated Nitrogen Carbon Analyser (ANCA). Stable C isotope values were expressed as $\delta^{13}C$ in per mil (‰) and normalised against the Pee Dee Belemnite (PDB) standard.

Precision and accuracy of the isotope ratios and C masses were assessed by repeated analysis of in-house standards (Isoleucine, *National Institute of Standards & Technology*, RM 8573, USGS40) with known isotopic composition. The precision of $\delta^{13}C$

measurements was better than ±0.2 ‰, the mean uncertainty for POC was ±3.34 µg, and concentrations were determined by dividing the POC content per filter by the volume of water filtered through that filter.

The relative organic carbon (OC) content of the TSM ($OC_{TSM}$, wt%) was calculated by dividing the sample POC content by the TSM content (Eq. 1)

$$OC_{TSM} = \frac{POC}{TSM} \cdot 100\% \tag{1}$$






Radiocarbon dating by accelerator mass spectrometer (AMS) was achieved on a Mini Carbon Dating System (MICADAS) following Mollenhauer et al. (2021). We report radiocarbon data as $\Delta^{14}$C values in ‰, which expresses the relative difference in $^{14}$C activity between the absolute international standard (reference year 1950) and the sample activity corrected for sampling time and normalised to $\delta^{13}$C=25 ‰ (Stuiver & Polach, 1977). The processing blank was determined by five empty combusted

blank filters (GF/F, 2.5 cm Ø) treated identically to the samples (Mollenhauer et al., 2021).

Since radiocarbon analysis is commonly used as a method for determining OM age, for discussion of the results, we refer to more $\Delta^{14}$C-depleted samples as "old" C from "old" OM sources and more $\Delta^{14}$C enriched as "young" or "modern" C.

### 2.3.3 Data representation and calculations

The data used in this study were subdivided into three groups according to the sampling locations and differences in the

hydrological regime and studied parameters: 1) samples from the Lena River main channel along the sampling transect from Yakutsk to the Lena Delta, 2) samples from near Stolb Island at the apex of the Lena River main stem and Lena Delta, and 3) samples from the Sardakhskaya branch in the Lena Delta itself. We discuss our data in the context of a compilation of summer data obtained by the ArcticGRO initiative (group 4). This dataset includes parameters measured from 2004 to 2019 (Holmes et al., 2021) in the summer between 15 June and 31 August. This subset of ArcticGRO samples was chosen to allow direct

comparison of the published results with data presented here and to avoid extreme events of the hydrologic system like the spring – early summer ice breakup (maximum water and suspended particulate matter [SPM] discharge) and winter ice-cover (minimum water and TSM discharge) (Magritsky et al., 2018).

The discharge data are provided by the Russian Federal Service for Hydrometeorology and Environmental Monitoring (Roshydromet) for the Lena River at Kusur (70.68°N, 127.39°E, see Figure 1a).

The groups of data defined for this study are described in figures 2, 3, 4 and 5 and table 2 as "Main Stem", "Delta", "Stolb" and "ArcticGRO".

Endmember modelling analysis was performed to derive quantitative estimates of the relative input of a potential C source endmember into the POC pool of every water sample and described in detail in 4.2.3.

## 3 Results

### 3.1 Depth distribution

We sampled water in the delta from different depths to investigate the distribution of OM through the water profile in the deltaic ecosystem. We did not observe systematic and significant differences between TSM, POC concentration, OC content, or carbon isotopes of POC for different water depths. Conductivity, Temperature and Depth (CTD) data measured during the sampling campaign showed no temperature or conductivity stratification of the water profile (Fuchs et al., 2022). Thus, to





characterise other studied parameters in the Lena Delta, we considered all the data measured in the delta from all water depths as one dataset without further subdividing the values into depth groups.

We did not collect samples from different water depths along the river transect from Yakutsk to Stolb but instead were only able to sample surface waters. In contrast to our surface water samples, ArcticGRO samples are depth-integrated. This difference in sampling might explain some of the differences between our observations and those made by ArcticGRO (see

sections 3.3.1 & 4.2.2).

## 3.2 Total suspended matter; particulate organic carbon and its content in total suspended matter

### 3.2.1 Total suspended matter

TSM concentration varied strongly between the river main stem with high spatial variability, and the Lena Delta, where TSM was distributed homogeneously (Figure 2a). The concentration of TSM from the main stem decreased from the upstream

catchment to the delta, from 34.5 to 15.0 mg L$^{-1}$, with an average of 22.7 ±6.3 mg L$^{-1}$ (mean ±stdev). We measured the highest TSM concentration at a station 150 km downstream from Yakutsk (WL19-02), where the largest Lena tributary (Aldan) flows into the main stem (Figure 1a; 2a). From there on, TSM concentrations consistently decreased, reaching a minimum at the two stations closest to the delta near Stolb Island (WL19-08 and -09).

Compared to the Lena River main stem transect, the TSM concentrations in the delta were lower and rather homogeneous,

displaying no obvious trends in spatial distribution. The highest TSM concentrations in the delta were found for several samples along the entire transect; 20.0 mg L$^{-1}$ 40 km downstream from Stolb (LEN19-S-03), 16.4 mg L$^{-1}$ in the middle of the Sardakhskaya branch (LEN19-S-05), and 19.4 mg L$^{-1}$ and 18.8 mg L$^{-1}$ closer to the delta outlet (LEN19-S-08 and -89). The mean concentration of TSM for the delta was 9.3 ±5.2 mg L$^{-1}$, which is lower than in the river main stem. The mean TSM concentrations at Stolb Island (transition zone between the Lena main stem and the delta) was 8.6 ±3.7 mg L$^{-1}$ with a maximum

up to 13.8 mg L$^{-1}$ in the middle of the profile , which is within the range of the lowest values measured in the main stem transect and the average deltaic values (Figure 1a).

TSM reported in the ArcticGRO dataset varied between 7.6 and 51.0 mg L$^{-1}$ with an average of 27.8 ±11.3 mg L$^{-1}$ and was similar to our main stem sample results.

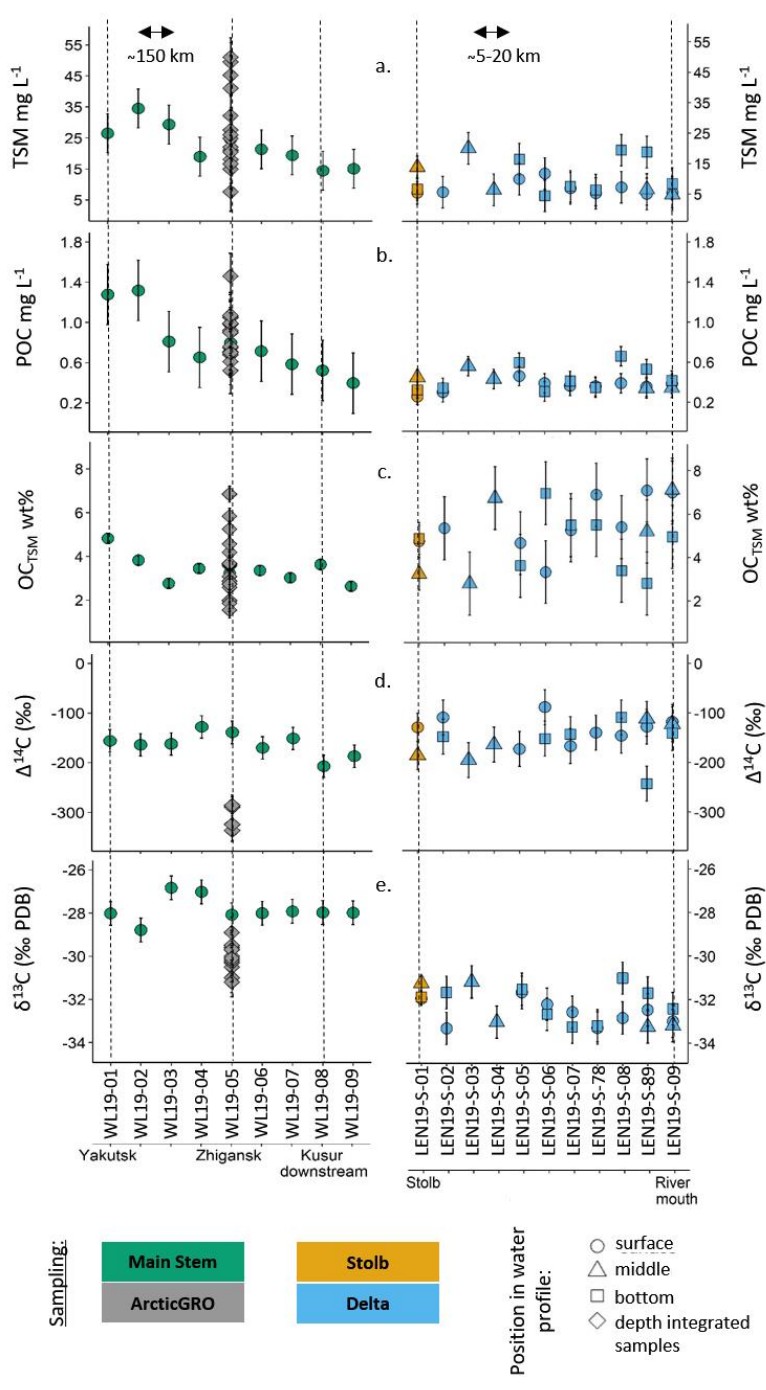

**Figure 2.** Distribution of studied parameters along the transect in the Lena main stem, the Lena Delta, and for the ArcticGRO dataset (mean ±stdev): a. TSM mg L$^{-1}$; b. POC, mg L$^{-1}$; c) OC$_{TSM}$, wt%; d) $\Delta^{14}$C of POC, ‰; e) $\delta^{13}$C of POC, ‰.



### 3.2.2 Particulate organic carbon and its content in total suspended matter

The distribution pattern of POC concentrations along the Lena River main stem was similar to that of TSM concentrations (Figure 2b). POC decreased by ~70 % downstream, from 1.32 mg L$^{-1}$ at Yakutsk (WL19-01) to 0.40 mg L$^{-1}$ at the last station
(WL19-09). The mean POC concentration in the Lena River was 0.79 ±0.30 mg L$^{-1}$. The highest POC concentration was measured at the two southernmost sites (WL19-01, WL19-02) (1.32 and 1.28 mg L$^{-1}$., respectively).

Deltaic POC concentration was on average lower than riverine POC with a mean of 0.41 ±0.10 mg L$^{-1}$ (range 0.26-0.66 mg L$^{-1}$).The samples which were characterised by high TSM had high POC concentrations as well. These values were similar to the range of POC concentrations found for the second half of the river main channel transect.

Like TSM, POC concentrations provided by the ArcticGRO database were very similar to the values we obtained in this study. The average POC concentration from ArcticGRO is 0.86 ±0.41 mg L$^{-1}$, with the highest values being 0.98 and 1.46 mg L$^{-1}$.

The POC content in TSM (OC$_{TSM,}$ wt%) from main stem samples was relatively constant along the complete river transect with a mean OC$_{TSM}$ of 3.4 ±0.2 wt%. The southernmost station (WL19-01; Yakutsk) was an exception, being enriched in suspended matter and especially in POC (TSM: 26.5 mg L$^{-1}$ and POC: 1.28 mg L$^{-1}$) and OC$_{TSM}$ was slightly higher than the
transect average (4.8 wt%) (Figure 2c). In the deltaic waters, OC$_{TSM}$ was significantly higher than in main stem samples, and the values within the delta increased from Stolb Island (4.3 ±0.7 wt%; LEN19-S-01) toward the river mouth (up to 7.1 wt% for LEN19-S-09, sampled at 5 m depth in delta disembogue). The average OC$_{TSM}$ in the delta was 5.2 ±1.5 wt%, with a minimum as low as 2.8 w% for stations with the highest TSM and POC concentrations (40 km downstream from Stolb, LEN19-S-03).

OC$_{TSM}$ at Zhigansk from ArcticGRO was in a similar range as the values we found in the main Lena River channel. The average value was 3.5 ±1.3 wt%. The two lowest values of 1.9 wt% were measured for samples with very high TSM and POC concentrations. Two high values stood out from the other values and were measured in the sample with the lowest POC and TSM concentrations. The mean OC$_{TSM}$ within ArcticGRO was not significantly different from the mean POC content in the river transect data, with values of 3.6 % and 3.4 %, respectively.

### 3.3 Isotopic composition of particulate organic carbon

### 3.3.1 Δ $^{14}$C of particulate organic carbon

Radiocarbon levels of POC varied within a wide range between -243 and -88 ‰ along the entire transect (Figure 2d), and did not differ substantially between river main stem and delta samples. The mean radiocarbon value for the main stem part of the transect was -163 ±23 ‰ (range from -207 to -128 ‰). The Δ$^{14}$C of POC at the first three upstream main stem stations (WL19-
01, -02, and -03) were similar at -160 ±3 ‰. Further downstream Δ$^{14}$C increased to -128 ‰ at the station ~ 150 km upstream Zhigansk (WL19-04) and decreased to -207 ‰ at station WL19-08 on the way to Stolb Island (LEN19-S-01), where it was -158 ‰. In the delta, the mean Δ$^{14}$C ‰ was -144 ±35 ‰ with a minimum of -243 ‰ at station LEN19-S-89 before the delta





disembogues into the sea and a maximum of -88 ‰ at LEN19-S-06 in the middle of the delta transect. Thus, Lena Delta POC in 2019 was within the same age as, or younger (higher $\Delta^{14}$C concentration) than POC in the Lena River main stem.

ArcticGRO reported significantly lower $\Delta^{14}$C values than in this study. The average $\Delta^{14}$C value for the ArcticGRO dataset was -309 ‰, while our value for Zhigansk in 2019 was -164 ‰ (WL19-05), which was similar to the average for the entire transect in 2019 (-163 ±23 ‰). Thus, radiocarbon values reported by ArcticGRO were depleted in $^{14}$C and differed from the values we measured in the Lena main stem and Lena Delta.

### 3.3.2 $\delta^{13}$C of particulate organic carbon

A strong difference in $\delta^{13}$C between the studied areas was revealed. $\delta^{13}$C values in the main stem were significantly higher than in the delta and at Stolb, with means of -27.9 ±0.6 ‰ in the main stem and -31.7 ‰ ±0.3 and -32.5 ±0.7 ‰ at Stolb and in the delta, respectively (Figure 2e). Values within the main stem part of the transect ranged between -28.8 and -26.8 ‰, with higher values for stations WL19-03 (-26.8 ‰) and WL19-04 (-27.0 ‰) after the Aldan and Vilui Lena tributaries exit the Lena main stem. Stable C isotope composition of POC within the delta and at Stolb displayed slightly more spatial variation, with

values between -33.1 and -31.0 ‰ and no specific distribution trend; thus, the lowest value of -33.3 ‰ was measured at several deltaic stations (LEN19-S-02, LEN19-S-07, LEN19-S-78, and LEN19-S-89) (Figure 2e). The $\delta^{13}$C of POC reported by ArcticGRO was -30.0 ±0.7 ‰, which was in between the values measured by us in 2019 at the Lena main stem and delta.

## 4 Discussion

### 4.1 Organic carbon load in the Lena main stem and the Lena Delta

Our dataset of TSM and POC concentrations sampled in 2019 displayed generally higher and more variable values along the Lena River main stem than in the Delta, while the OC content of TSM was higher in the Lena Delta. This pattern could be explained by a) the local flow regime and its hydrological changes such as flow velocity, water and sources distribution, and mean catchment slope along the Lena main stem and its Delta, or b) the time of sampling relative to the annual spring-summer flooding and the discharge fluctuations, or c) a combination of both factors. In the following, we discuss these factors in detail.

### 4.1.1 Discharge and sampling time


The Lena River is characterised by a nival hydrograph regime with a distinct flood event taking place in the beginning of summer during the snowmelt and ice breakup period (May-June) and a very low water flow in winter (Yang et al., 2002) which is typical for the north-flowing East Siberian rivers. Discharge has a strong effect on the amount of solids and OM released by a river (Magritsky et al., 2018). The peak of annual POC concentrations (>3.6 mg L$^{-1}$, McClelland et al., 2016) and TSM

concentrations (>150 mg L$^{-1}$, ArcticGRO) occur right after the flooding following ice breakup in late May–early June. The peak water yield in 2019 took place on 2 June and reached 83,000 m$^3$ s$^{-1}$, then it decreased and varied in the range of 49,200–45,999 m$^3$ s$^{-1}$ during the time interval when the main stem transect was sampled. During the sampling campaign in the





Lena Delta (2019/08/07–2019/08/09) the Lena River discharge was 19,600–19,000 m$^3$ s$^{-1}$, which was less than half the discharge during main stem sampling. In comparison, typical winter discharge is around 2,000 m$^3$ s$^{-1}$ (average winter minimum

discharge in 1980-2014 by Magritsky at al., 2018).

We analysed all ArcticGRO data on TSM and POC for the Lena River comprising samples collected in the years 2003-2021, demonstrating that TSM and POC concentrations were correlated with discharge (Figure S1). However, when considering a discharge ranging between 15,000 and 50,000 m$^3$ s$^{-1}$, corresponding to the typical range of discharge values in summer and in early autumn (September), as well as to the range of discharge values observed during our two sampling campaigns, there is

no significant relationship between discharge and TSM and POC (Figure S1). Thus, the strong relationship that has been described to persist between discharge and TSM appears to be driven by the large difference between maxima in all parameters (observed during the spring flood) and their minima (found during low flow in winter). The discharge observed during both sampling campaigns in 2019 varies within the range for which low impact on suspended matter load was found.

Average surface water TSM and POC concentrations in the Lena River in 2019 agree with reported average TSM and POC

concentrations observed by ArcticGRO during periods with discharge within the above mentioned range (TSM in this study; 21.29 mg L$^{-1}$ as compared with 22.66 mg L$^{-1}$ for ArcticGRO, and POC 0.77 mg L$^{-1}$ and 0.79 mg L$^{-1}$, respectively). Thus, 2019 could be characterised as a year with "representative" to "lower-than-average" TSM exports. Such fluctuations between years have been observed before (Kutscher et al., 2017). Kutscher et al. (2017) reported POC concentrations in the Lena main stem in June 2013 (mean 0.38 mg L$^{-1}$) which were even lower than POC in the river main stem found in this study for June 2019

(Figure 3).

TSM and POC concentrations in the Lena Delta in summer 2019 were 2 and 1.5 times lower, respectively, than values reported by ArcticGRO for a comparable time of year and under similar discharge conditions (TSM: 9.3 ±5.2 mg L$^{-1}$, POC: 0.41 ±0.10 mg L$^{-1}$) (Figure 2a & b). Our data from Stolb likewise show lower values than those from ArcticGRO (TSM: 8.6 ±3.7 mg L$^{-1}$, POC: 0.34 ±0.10 mg L$^{-1}$). On the other hand, our deltaic POC concentrations are similar to previously published POC data for

the Lena Delta (Winterfeld et al., 2015) (Figure 3). This shows that the difference we observe between river and delta is a persistent feature that is not biased by sampling time or depth but is mostly caused by other factors such as, e.g., flow and velocity.

### 4.1.2 Hydrology of the Lena River

The Lena River watershed can be subdivided into several areas, which contribute differently to the TSM and water discharge

into the Lena and are characterised by distinct morphologies. The separation between the Upper and Lower Lena takes place approximately at Yakutsk. The Upper Lena includes the southern limits of the river. Its watershed covers an extensive area between Lake Baikal and Yakutsk and includes dozens of tributaries including creeks and small rivers. The Lower Lena flows from Yakutsk into the Laptev Sea and receives waters from catchments including the Verkchoyansk Range. Thus, the first station of our main stem transect was situated at the Upper Lena and the second station marks the transition from Upper to

Lower Lena. According to Kutscher et al. (2017), Upper and Lower areas of the Lena account for the following fractions of





the total watershed area and POC discharge: Upper Lena (37 % of the entire Lena River watershed, 2.9 Tg C yr$^{-1}$ of TSM), Aldan, the watershed of the Lena River's main tributary (29 %, 1.8 Tg C yr$^{-1}$), Lower Lena (15 %, 1.1 Tg C yr$^{-1}$) and Vilui, the second big tributary of the Lena (19 %, 0.6 Tg C yr$^{-1}$).

Surface river water concentrations of TSM and POC in 2019 display a decreasing trend from Yakutsk to Kusur downstream
along the Lena River (Figure 2a & b). The highest TSM and POC concentrations (34.5 mg L$^{-1}$ and 1.32 mg L$^{-1}$, respectively) were found at the location close to where the Aldan tributary flows into the Lena River (WL19-02); downstream from where the Vilui tributary disembogues into the Lena (WL19-03), TSM and POC concentrations steadily decrease. Several factors may account for this observation.

OM concentrations in the Lower Lena were lower compared to the Upper Lena. It has been suggested that this is related to the
difference in the geological setting in the northern part of the catchment compared to the southern (Kutscher et al., 2017) and to the mean slope of the Upper Lena subcatchment, which correlates with OM concentrations in the river (Mulholland, 1997). The Upper Lena is sourced from regions with higher precipitation (Chevychelov & Bosikov, 2010), smaller extent of continuous permafrost (Obu et al., 2019), and more productive forests (Stone & Schlesinger, 2003). Waters entering the Upper Lena carry OM supplied by tributaries from the Central Siberian Plateau with waters from the relatively flat Aldan watershed.
In contrast, the Lower Lena receives a considerable part of the water flowing from the steeply sloped mountainous areas of the Verkhoyansk Range. The water catchment area of the Lower Lena is covered by shallow, OM-poor soils: Gleyzems, which are Al-Fe-humic soils, and shallow, weakly developed soils that develop in mountainous areas (Stolbovoi, 2002). OM from the Upper Lena catchment may reach the Lower Lena, as we measured at the beginning of the riverine transect. TSM and POC concentrations decreased in the Lower Lena regime because local tributaries carry less OM, with possibly relatively more
mineral compounds, which, for example, is reflected in the decrease in OC$_{TSM}$ from the south to the north of the main stem transect (Figure 2c).

The lower TSM and POC downstream from the Vilui could also result from the hydrological change, which takes place inside the Lower Lena itself. Kääb et al. (2013) reported that the velocity of the Lena River decreases downstream from south to north along the 620 km long Lena River transect between 67.00° and 71.58°N (corresponding to the river stretch between our
stations WL19-05 [Zhigansk] and WL19-09 [the nearest to the delta]), reaching the lowest values approximately 40 km south of Kusur. This is a result of the topography of the region, where the base slope elevation of the river flattens near 150 km south of Kusur. It is also known that the majority of TSM brought by the Lena River from the water catchment is deposited before the Lena reaches the stream north of Kusur, which is known as the "Lenskaya Truba" (which means "Lena pipe"). This is the narrowest (sometimes less than 2 km) and deepest (more than 20 m) part of the river (Fedorova et al., 2015) where the TSM
sedimentation takes place. The preferential sedimentation of mineral particles may also affect the isotopic composition of transporting OM. As it was shown by Vonk (2014) contemporary terrestrial OM is dispersed mainly by horizontal transport, while mineral-associated (i.e., heavier), older OC from topsoil and Yedoma is affected mainly by vertical transport and seem to settle rapidly. Thus, decreasing velocity in the Lower Lena and the Lena Delta allows further sedimentation of TSM and old C, resulting in a decrease of its concentration.





The Lena Delta was characterised by the lowest TSM and POC values in 2019, as previously described. Several factors may account for these findings. It is known that the sediment transport from the Lena Delta to the Laptev Sea is controlled by the distribution of water and sediment discharge along the different deltaic branches (Rachold et al., 1996). According to Rachold et al. (1996), the Trofimovskaya-Sardakhskaya branch system receives 61 % of the annual water volume registered at Stolb station. These channels have complex structures and contain several different regional sub-delta systems. As a result, only 7

to 8 % of the initial water volume reaches the mouth of the Trofimovskaya channel (Rachold et al., 1996). The water, loaded with suspended matter, is distributed into the numerous channels; this affects the amount of TSM detected in the big channels. Due to the extensive branching of the channels in the delta velocity decreases, allowing for settling and sedimentation of TSM, mostly on flood plains (Sanders et al., 2022) following the high-water flooding season.

        The OM$_{TSM}$ is much higher in the delta compared to the Lena main stem, which may be a result of preferential settling of

denser mineral-rich particles between the main stem and the delta. Further input of OM originates from the islands, supplied by bank erosion and degradation of ice-rich permafrost deposits such as Yedoma; both processes take place in the delta (Stettner et al., 2018; Fuchs et al., 2020; Haugk et al., 2022). Based on these observed patterns, we suggest that the Lena Delta should be distinguished as the third hydrographically and sedimentologically distinct part of the Lena River system.

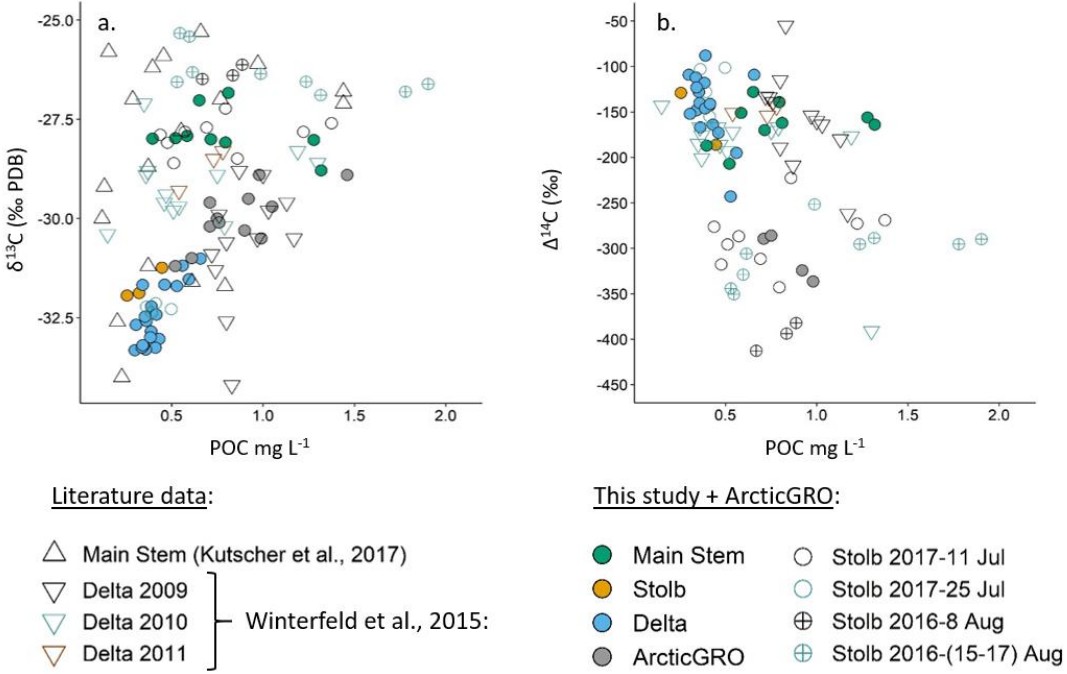

**Figure 3.** Stable and radiocarbon isotopic values and POC concentration for the Lena Delta and the Lena River main stem. Values were taken from this study and from the literature (Kutscher et al., 2017 and Winterfeld et al., 2015).





### 4.2 Potential sources of particulate organic carbon

#### 4.2.1 Stable carbon isotope sources

The $\delta^{13}C$ values of POC in the Lena main stem surface water ranged between -28.8 and -26.8 ‰ with a mean of -27.9 ±0.6 ‰,
comparable to a typical $\delta^{13}C$ of terrestrial C3 plants (about -27 ‰) (Finlay & Kendall, 2008). C3 plants are dominant in the
Lena catchment area and account for most of the soil OM (SOM) (from -28.4 to -27.0 ‰), including that of Yedoma deposits
(-27.5 ‰) that have been described for the Lena watershed (Vonk et al., 2017, Schirrmeister et al., 2011), and for the terrestrial
primary production (-27.7 ‰) such as organic and litter layers (Wild et al., 2019). The observed $\delta^{13}C$ values also overlap with
the range of $\delta^{13}C$ values determined for soils from the first terrace of Lena Delta soils (from -27.0 to -25.1 ‰) (Winterfeld et
al., 2015).

In contrast to main stem POC, deltaic POC was more depleted in $\delta^{13}C$ (mean -32.5 ±0.7 ‰) (Figure 2e; 3c). Similarly low
values were found in the delta in 2016 and 2017 and were previously reported in several publications for the Lena Delta (e.g.
Winterfeld et al., 2015), the Lena River (Kutscher et al., 2017) (Figure 3), and the Kolyma River (Bröder et al., 2020). Those
authors related depleted riverine POC to the influence of aquatic primary production and suggested a phytoplankton
contribution to explain the isotopic composition of POM. However, in these studies reported POC $\delta^{13}C$ values below -30 ‰
were the exception rather than the rule. This is in contrast to our set of deltaic samples from 2019, where POC is consistently
strongly depleted in $\delta^{13}C$, showing $\delta^{13}C$ values between -33.3 and -31 ‰.

Unfortunately, neither of the cited publications reports the phytoplankton biomass and/or chlorophyll-*a* data for the studied
water samples, nor were these measurements part of our sampling campaign. This lack of quantitative observational data
requires a theoretical estimation of phytoplankton input into POC.

According to data from Finlay and Kendal (2008), $\delta^{13}C$ from phytoplankton in river POC ranges from -42 to -25 ‰. However,
these authors stress that the range of isotopic values is smaller within each single river. Regardless, depleted $\delta^{13}C$ values in
POC, as observed in our deltaic samples, are consistent with substantial contributions of POC from aquatic production during
phytoplankton bloom.

In the Arctic, 2019 was a warm and dry year when mean annual surface air temperature over the Arctic land mass was the
second highest in the observational period (1900–present) (Richter-Menge et al., 2019). Our Lena Delta sampling period was
between 20 July and 10 August, which is the season when the highest water temperatures were observed by Liu et al. (2005).
Higher water temperatures induce higher metabolic rates and thus increased aquatic primary production as well as
heterotrophic activity (Paczkowska et al., 2019; Bosco-Galazzo et al., 2018; Allen et al., 2005). The average water temperature
in the delta in 2019 according to the CTD measurements was 15.2 °C (Fuchs et al., 2022). Liu et al. (2005) reported a decrease
of water temperatures from the Upper Lena to the north. Average Lena water temperatures at Kusur (the northernmost station
of that study) were 3-5 °C lower in mid-open water season (end of June-end of August) compared to the Aldan and Upper
Lena basins and barely reached 15 °C (Liu et al., 2005). Likewise, ArcticGRO reports a water temperature of 15.0 °C on 2
August 2019. Thus, the water temperatures in the Lena Delta during our sampling campaign (7-8 August) were higher than in





Zhigansk (~800 km to the south of the delta) despite the latitudinal gradient in water temperatures observed by Liu et al. (2005). A heat wave in 2019 may have had further consequences in the Lena Delta, such as the formation of isolated small deltaic channels, which were separated due to extremely low water conditions. These channels may represent favourable habitats for the development of high numbers of algae despite the absence of an additional nutrient flux, as temperature and velocity appear to exert a stronger positive influence on phytoplankton growth, outbalancing the lack of nutrients (Li et al., 2013). From this,

it follows that the delta provided more favourable thermal conditions for phytoplankton than did the main stem; this was particularly true in the very warm year of 2019. We suggest that sampling during potential algal blooms took place in earlier years as well. For example, in 2017 we observed a $\delta^{13}$C decrease from -27.9 ‰ to -32.3 ‰ within only two weeks (11 July – 25 July 2017), suggesting that an algal bloom developed between the two sampling dates (Figure 3c).

Additional support for the increase of aquatic production in August 2019 is provided by gradients in nitrate (1.5 µmol L$^{-1}$ to

0.25 µmol L$^{-1}$) and silicate (50 µmol L$^{-1}$ to 18 µmol L$^{-1}$) observed by Sanders et al. (2022) on the way along the sampling transect from Stolb Island further into the delta (LEN19-S-01, -02 and - 03). Sanders et al. (2022) suggest that phytoplankton such as diatoms may be responsible for this uptake, as described in Hawkings et al. (2017).

The C isotopic fractionation during photosynthesis is not constant and may depend on the environmental conditions, but isotopic composition of phytoplankton generally strongly correlates with the isotopic composition of dissolved inorganic C

(DIC) (Rounick and James, 1984). In the Lena Delta the $\delta^{13}$C of DIC has not been determined, but the low $\delta^{13}$C of POC suggests a $^{13}$C-depleted DIC pool. Low $\delta^{13}$C in DIC can be caused by several processes as shown by Brunet (2005) for riverine DIC in the rivers of Patagonia: degradation of dissolved OC (DOC) containing soil organic C will result in low $\delta^{13}$C of DIC, and $\delta^{13}$C of DIC was found to be negatively correlated with DOC concentration. It is conceivable that DOC remineralisation leads to similarly reduced $\delta^{13}$C of DIC in the Lena Delta, which is in agreement with observations of rapid remineralisation of

DOC released from thawing permafrost as reported for the Kolyma catchment (Mann et al., 2015) and for thaw creeks on an island in the Lena Delta (Stolpmann et al., 2022).

The fact that the lowest $\delta^{13}$C values tend to be in samples with the lowest POC concentrations offers another interesting perspective (Figure 3a). As discussed above, TSM content (and POC from terrestrial sources) appears to be related to flow velocity. It is known that velocity exerts a strong negative effect on Chlorophyll-*a* concentration (Li et al., 2013). Together

with other hydrological properties such as increasing suspended matter and turbidity, velocity could be one of the critical forcing factors regulating phytoplankton biomass, high velocity dilutes phytoplankton cells, reduces light availability, and changes the entire dynamic of aquatic production (Salmaso & Braioni, 2008). In turn, low flow velocities in the deltaic channels, as suggested by low POC values, could provide more favourable conditions for aquatic production while at the same time, larger or denser mineral-bearing particles might be settling.

The combined factors of low flow velocity in the shallow delta channels, where sunlight penetrates much of the water column which contains only small amounts of suspended particles, and the extremely warm summer conditions during our sampling campaign might have resulted in high primary production providing larger relative amounts of POC than in the main stem and during previous years.





$\delta^{13}$C from ArcticGRO was lower than the $\delta^{13}$C we measured in the main stem, which may indicate more phytoplankton
contribution to ArcticGRO samples. This may have resulted from potential higher C3 plant contribution to POC in the main
stem in 2019 instead of contributions from phytoplankton, which was very likely higher for ArcticGRO samples due to the
sampling time (9 of 12 $\delta^{13}$C ArcticGRO records were measured in samples taken in August and at the end of July), and/or
from less DOC recycling in June (main stem sampling 2019) than at the end of July –August (ArcticGRO).

### 4.2.2 $\Delta$ $^{14}$C in particulate organic matter

$\Delta^{14}$C of POC sampled in 2019 was homogeneous along the studied transect, both along the Lena River main stem and within
the Lena Delta (Figure 2d). In contrast to this, $\Delta^{14}$C values reported by ArcticGRO were strongly depleted in $^{14}$C compared to
our 2019 transect data, although the values found in this study for the Lena Delta and nearby Stolb Island fit within the range
of other previously published data from the Lena Delta (Figure 3b; 4). Values published by Winterfeld et al. (2015) for multiple
summer seasons in the Lena Delta vary between -145 and -194 ‰; Karlsson et al. (2016) reported values in the range of -433
to -97 ‰ (Figure 4). This comparison reveals that the 2019 deltaic transect data are not atypical, as they are similar to the
values for August 2009 and very close to July-August 2010 and June-July 2011.
The $\Delta^{14}$C of POC from Stolb in 2016 and 2017 were relatively similar to ArcticGRO values or sometimes lower (Figure 3b;
4). It is important to highlight that $\Delta^{14}$C in POC may change strongly within a relatively short period. For example, sampling
in the summers of 2016 and 2017 was done twice with only one (2016) or two (2017) weeks in between. $\Delta^{14}$C in 2017 increased
over the course of two weeks (11-25 July) by almost 170 ‰ causing values to jump from -288 ‰ to -122 ‰ (Figure 3b). The
data from 2016 also showed a distinct but less pronounced increase in $\Delta^{14}$C within one weeks (8-15 August) from -387 ‰ up
to -306 ‰. Thus, $\Delta^{14}$C of POC is a parameter that exhibits large variability along the Lena main stem and inside its Delta. This
variability makes this parameter hard to interpret and suggests that local and short-term effects exert strong control (e.g.,
collapse of a bluff along the channel bank, or rain events).
The difference between ArcticGRO and the 2019 data may also be explained by the rather small number of ArcticGRO
observations used for this comparison. The mean $\Delta^{14}$C value of the entire Lena dataset from ArcticGRO is -275 ‰ and ranges
from -398 to -164 ‰, which includes our $\Delta^{14}$C values observed in 2019. We did not consider the entire ArcticGRO dataset for
Lena River samples as we expect that during the discharge peak following the ice breakup in spring, POC might be of distinctly
different origin than during the summer. We thus selected only values obtained for samples collected in the summer season
after 15 June until 31 August. As a result, the reduced ArcticGRO dataset consists of only four samples with a mean $\Delta^{14}$C
value of -309 ‰ (min/max -336/-286 ‰) sampled in August of 2004 and 2005, similar to the time of the year when our deltaic
campaign took place.
Another explanation for the difference in $\Delta^{14}$C of POC between ArcticGRO and our riverine transect may be the fact that
ArcticGRO samples are depth-integrated, while our samples from the Lena River in 2019 are surface water samples, and from
discrete water depths for the samples from the Lena Delta. The Lena River itself is deep along its main stem (up to 20 m





downstream from Yakutsk); thus, pronounced and systematic differences between water mass properties like temperature, light penetration, etc. could prevail, in turn influencing the composition of POC. We speculate that surface water POM might be biased towards more phytoplankton contribution in the sunlit surface waters and/or toward the contribution of vascular plants floating in the surface waters of the stream, resulting in apparently younger POM than at depth. This might explain why

our 2019 river samples are systematically younger than the ArcticGRO values.

Samples with lower POC concentrations tend to display higher $\Delta^{14}C$ values and vice versa (Figure 3b). We suggests that conditions with low suspended particle load promote phytoplankton growth, while highly turbid waters contain substantial amounts of POC derived from riverbank erosion or cliff failure.



**Figure 4.** The origin of the POC in the Lena River and the Lena Delta. Endmember values (red crosses) were combined after Galimov et al., 2006, Wild et al., 2019, and Winterfeld et al., 2015; additional previously published data were added from Winterfeld et al., 2015 and Karlsson et al., 2016.



### 4.2.3 Estimation of organic matter sources based on endmember modelling

Combined radiocarbon and stable C isotope compositions are used to investigate the origin of OM. The difference in OM
origin for POC measured along the transects in 2019 was illustrated by stable C isotope variations. $\delta^{13}C$ values in the delta
suggest an aquatic origin of POC, and those samples also display comparatively high $\Delta^{14}C$ values. On the other hand, $\Delta^{14}C$ of
POC in the delta and in the main stem are similar, but differ in $\delta^{13}C$ values from the river samples, which are less depleted in
$\delta^{13}C$. If the delta samples contain substantial amounts of aquatically produced OM, isotope mass balance considerations require
that some of the POC collected from delta waters still derive from terrestrial sources which are older than the sources supplying
POC to the main stem. To examine this hypothesis, we applied an endmember model.

To illustrate possible sources of OM, we used a dual-carbon-isotope ($\Delta^{14}C$, $\delta^{13}C$) three-endmember mixing model. Riverine
POM is regarded as originating, to variable degrees, from autochthonous production of phytoplankton and from allochthonous
sources such as vegetation and soils (Ittekkot & Laane, 1991). Terrestrial allochthonous sources for the Lena River catchment
may be divided into two pools with different ages and origins: Holocene permafrost soils and organic-rich Pleistocene deposits.
Endmembers for the OM sources in the Lena main stem and its Delta were defined as phytoplankton (I), Holocene soils (II),
and Pleistocene deposits (III), and the respective endmember isotope values were taken from previously published studies
(Table 1). For Holocene soils, endmember values were taken from Winterfeld et al. (2015). This study focused on the Lena
Delta, and Holocene soil endmembers were combined from data measured directly for deltaic soils by this group of authors in
2009, 2010, 2011 and from a literature review. $\delta^{13}C$ values which we chose to use as endmember values were measured for
Holocene soils which resemble soils in Lena River ecosystems, for example from the Yenisey watershed, and $\Delta^{14}C$ values
were measured mostly within the Lena Delta in different years (Winterfeld et al., 2015).

Endmember values for ancient permafrost Yedoma have also been measured. We use endmember values from Wild et al.
(2019). In this study $\Delta^{14}C$ and $\delta^{13}C$ values of Pleistocene deposits were constrained using observations from Yedoma deposits
from multiple researchers which were combined together for, in total, 329 observations for $\Delta^{14}C$ and 374 for $\delta^{13}C$.
The determination of phytoplankton endmembers required further consideration. The $\delta^{13}C$ endmember values for
phytoplankton assumed in previous studies (e.g., Mann et al., 2015; Vonk et al., 2010; Wild et al., 2019; Winterfeld et al.,
2015) are higher than the range of $\delta^{13}C$ measured by us in the 2019 samples. The $\delta^{13}C$ values reported by Galimov et al. (2006)
for phytoplankton, however, are in the same range as ours from the Lena Delta and Stolb Island (-32.4 ‰). Those authors
determined $\delta^{13}C$ values of phytoplankton in the Yenisey estuary ranging from -27.8 to -37.0 ‰ , where phytoplankton is
dominated by species of the phylum Bacillariophyta, which is the dominant phylum in the Lena Delta as well (42.3 % of all
the species; Gabyshev et al., 2019). Therefore, we use the endmember value of -33.3 ±2.3 ‰ for $\delta^{13}C$ of phytoplankton.

We determined the $\Delta^{14}C$ endmember value for phytoplankton based on the assumption that recent terrestrial and aquatic
vegetation contains mostly modern C from the atmosphere, potentially even carrying elevated levels of $^{14}C$ affected by nuclear
weapons testing during the 1960s and 1970s; thus, values from organic litter from Russia, Scandinavia, and Alaska were
included (Wild et al., 2019).





**Table 1.** Endmembers for C isotope composition used in this paper (after Winterfeld at al., 2015, Wild et al., 2019, and Galimov at al., 2006).

| Endmember: | $\delta^{13}$C (‰) | $\pm\delta^{13}$C | Source | $\Delta^{14}$C (‰) | $\pm\Delta^{14}$C | Source |
|---|---|---|---|---|---|---|
| Phytoplankton | -33.3 | 2.3 | Galimov at al., 2006 | 97 | 125 | Wild et al., 2019 |
| Holocene soils | -26.6 | 1.0 | Winterfeld at al., 2015 | -282 | 133 | Winterfeld at al., 2015 |
| Pleistocene deposits | -26.3 | 0.7 | Wild et al., 2019 | -955 | 66 | Wild et al., 2019 |

Isotope mass balance endmember modelling was based on the following equations:

$$f_{\text{phytoplankton}} + f_{\text{holocene soils}} + f_{\text{pleistocene deposit}} = 1 \qquad (2)$$

$$\delta^{13}C_{\text{sample}} = f_{\text{phytoplankton}}\, \delta^{13}C_{\text{phytoplankton}} + f_{\text{holocene soils}}\, \delta^{13}C_{\text{holocene soils}} + f_{\text{pleistocene deposit}}\delta^{13}C_{\text{Pleistocene deposit}} \qquad (3)$$

$$\Delta^{14}C_{\text{sample}} = f_{\text{phytoplankton}}\, \Delta^{14}C_{\text{phytoplankton}} + f_{\text{holocene soils}}\, \Delta^{14}C_{\text{holocene soils}} + f_{\text{pleistocene deposit}}\Delta^{14}C_{\text{Pleistocene deposit}} \qquad (4)$$

where $f_{\text{phytoplankton}}$ , $f_{\text{holocene soils}}$, and $f_{\text{pleistocene deposit}}$ are the fractions contributed to the samples by phytoplankton, Holocene soils, and Pleistocene deposits, respectively. We applied the random sampling computer simulation (Monte Carlo simulation), which is based on the assumption that the endmember values are represented by a normal distribution, where the mean and a standard deviation were taken from previously published studies (Galimov et al., 2006; Wild et al., 2019; Winterfeld et al., 2015). Calculations were conducted using random sampling from this distribution while simultaneously applying the random sample

to mass balance equations (Eq., 2, 3, 4) with repeated random sampling (5,000,000 times). Modelling was carried out in R studio using the code published by Andersson (2011) in a modified version (Grotheer et al., 2020).

Based on this endmember model, the phytoplankton contribution is highest in deltaic samples with a mean of 68 ±6 % (Table 2, Table S1 and Figure 4; 5), whereas main stem POC consists of approximately 39 ±8 % aquatically produced material. In contrast, Holocene soils account for 13 ±10 % in the delta versus 56 ±12 % in riverine POC. Moreover, dual-carbon-isotope

($\Delta^{14}$C, $\delta^{13}$C) three-endmember mixing model results are consistent with input from an additional old permafrost source into deltaic POC. Thus, the Pleistocene contribution to the Lena Delta POC was estimated as 18 ±4 % which is almost four times higher compared to the main stem POC which held less than 5 % of ancient permafrost C in 2019 (Table 2, Table S1, and Figure 5); this demonstrates an additional source of permafrost derived C, particularly in deltaic waters.

Our estimation of permafrost input into POC in the Lena main stem differs from that of Wild et al. (2019) who estimated pre-

aged material input to the Lena River in summer as 62 ± 0.5 %. Such a large variation is explained by the different approaches applied. Wild et al. (2019) divided endmember sources into 1) recent terrestrial primary production (vegetation, phytoplankton), and 2) pre-aged OM, which included all terrestrial OM: active layer, Holocene peat and thermokarst Pleistocene deposits. Phytoplankton was not included as a potential source, but it is likely included in the recent primary production category. In our study, we did not consider Holocene soils to be ancient C, but emphasised Pleistocene deposits





(Yedoma) as the contribution of ancient C in riverine and deltaic POC. The combined contribution to main stem POC from both Holocene soils and Pleistocene deposits was estimated by us to be ~ 61 % , which is similar to the contribution from a pre-aged OM source measured by Wild et al. (2019).

**Table 2.** Relative OM source contribution to water sample POC in 2019 (mean ±stdev).

|  | I. Phytoplankton, % | ±*stdev* | II. Holocene soils, % | ±*stdev* | III. Pleistocene deposit, % | ±*stdev* |
|---|---|---|---|---|---|---|
| Lena main stem | **39** | *8* | **56** | *12* | **5** | *4* |
| Delta* | **68** | *6* | **13** | *10* | **18** | *4* |

*For this model samples from Stolb Island were included with the deltaic samples and contributed to the final mean ±stdv values for potential POC sources.

In Figure 5, we show the estimated contributions of different OM sources to the actual average POC concentration measured for the Lena main stem and delta. This shows that POC derived from Holocene soils decreased from 0.44 to 0.05 mg L$^{-1}$ due

to sedimentation, which takes place in the Lower Lena, particularly downstream from the Kusur station, and in the delta itself as explained in section 4.1.2. POC derived from Pleistocene deposits almost doubles from 0.04 ±0.02 (main stem) to 0.07 ±0.02 mg L$^{-1}$ (Lena Delta). This is the case despite the lower concentration of POC in the delta than in the main stem and the lower discharge during the sampling time in the delta. This suggests that the Lena Delta receives POC from an additional ancient permafrost deposit source, which is specific to the Lena Delta. These observations support a finding from Karlsson et

al. (2016), who estimated the contribution of carbon derived from Pleistocene deposits to surface coastal surface sediments for the East Siberian Arctic Shelf to be ~53 % with surface soil contribution estimated to be only ~ 0.23 %. Thus, the type of material that reaches the sea floor in the coastal zone reflects Yedoma deltaic origin.





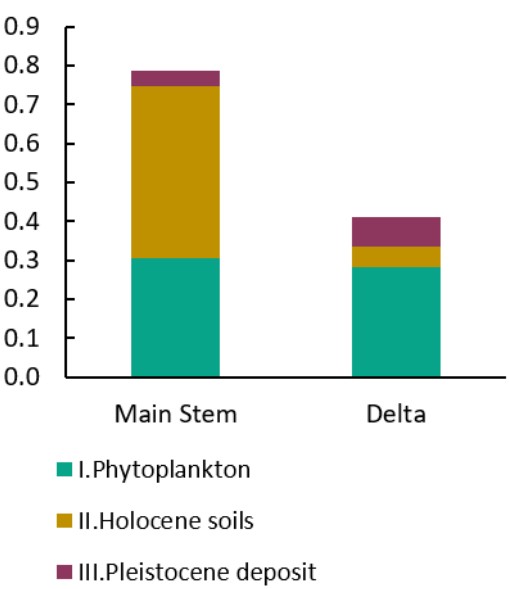

**Figure 5.** Contributions of different OM sources to the POC measured in the Lena River main stem and the Lena Delta along
the transect in 2019.

## 5 Summary and Conclusions

We found a significant difference between the two investigated parts of the Lena River system, the main stem and the delta.
TSM and POC in the main stem were significantly higher than in the delta. At the same time, TSM and POC concentrations
along the main stem remained within the range of values registered by ArcticGRO for the years 2009–2019, while deltaic
values of TSM and POC hardly reached the lowest values measured for Zhigansk. Conversely, the $OM_{TSM}$ is higher in the
delta than in the river, likely resulting from a different composition of the suspended matter.

The distribution pattern of TSM along the main stem decreases downstream to Stolb Island, which is the apex of the Lena
Delta. This suggests that the major enrichment in TSM (mass wise) of the Lena main stem takes place along the main channel
in the Upper Lena catchment. This TSM is enriched with mineral compounds, which tend to settle out on the way to the ocean
where river velocity decreases. Settling is even more pronounced in the delta, where flow velocity is lower because in the delta
the main stem divides into multiple branches.

Modern OM, possibly from phytoplankton primary production, dominates in the delta, combined with a higher input of ancient
permafrost, while riverine OM is predominantly derived from soil OM. The actual concentration of ancient permafrost-derived
POC in the delta exceeds the concentration in the main stem (~ 0.07 ±0.02 and 0.04 ±0.02 mg L$^{-1}$, respectively) despite the
lower concentration of POC (0.79 ±0.30 mg L$^{-1}$ in the river main stem and 0.41 ±0.10 mg L$^{-1}$ in the delta).

Our findings suggest that if an estimation of Lena OM discharge to the coastal zone is based only on the data from the Lena
main stem, it may overestimate the load of TSM and underestimate its sedimentation, which takes place in the Lower Lena



and its delta. In order to predict the effects on coastal waters of changes in permafrost due to climate change, additional and concurrent sampling from the river delta and the river main stem is needed. Particles that are mobilised from thawing

permafrost in the Lena catchment may be deposited and decomposed on the way to the ocean. Therefore, investigating permafrost fingerprints only in samples from the main stem may lead to incorrect conclusions and a biased perspective of permafrost carbon release to the coastal zone and the Arctic Ocean. The Lena Delta provides an additional source of permafrost carbon; Yedoma-derived OM, as part of the total permafrost carbon, could be then discharged into the Arctic coastal waters. The Lena Delta as the interface between the Lena River and the Arctic Ocean plays a crucial role in determining the qualitative

and quantitative composition of OM discharged into the Arctic Ocean.

*Data availability.* The data presented in this study will be freely available in the PANGAEA data repository and now are available upon request.

*Author contributions.* OO, GM, and JS designed the concept of the study. OO drafted a first version of the manuscript and carried out laboratory analyses of TSM and POC including radio C analyses together with HG. BJ designed the maps. HG provided R code for endmember analysis. TS, MF, JP, JS, and OO carried out the field work and collected the samples. All co-authors contributed to the manuscript writing and editing processes.

*Financial support.* This research is part of the 'Changing Arctic Carbon cycle in the COastal Ocean Near-shore (CACOON)' project funded by the Bundesministerium für Bildung und Forschung (grant no. 03F0806A) and the Natural Environment Research Council (grant no. NE/R012806/1).

*Competing Interests.* The authors declare that they have no conflict of interest.


*Acknowledgments.* We are grateful to everyone who helped and supported the CACOON project and the joint Russian-German "Lena 2019" expeditions, particularly Volkmar Aßmann (AWI) for logistics during the summer expedition and the Samoylov Research Station for hospitality. We would particularly like to thank Waldemar Schneider (AWI) for invaluable help with sampling along the Lena River main stem.

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
