# Peer review of "Particulate organic matter in the Lena River and its Delta: From the permafrost catchment to the Arctic Ocean"

_Biogeosciences, 2022_

## Referee Comment (RC3)

[referee-annotated manuscript omitted]

---

## Author Comment (AC1)

**Discussion on the 1ˢᵗ Anonymous Referee review**

Referee comment on "Particulate organic matter in the Lena River and its Delta: From the permafrost catchment to the Arctic Ocean" by Olga Ogneva et al., Biogeosciences Discuss., https://doi.org/10.5194/bg-2022-183-RC1, 2022

Comment types: Authors' Response: "AR", Referee Comment: "RC"

Comment colors: Authors Response: "blue", Referee Comment: "black"

Comment fonts: When it was possible, we highlighted changed text by the **bold font**, the text from the manuscript copied to this review was taped *cursive*

RC: Review of the manuscript "Particulate organic matter in the Lena River..." by Ogneva et al.

The paper addresses a fundamental question of riverine fluxes of particulate organic carbon in still poorly studied permafrost regions, and its potential impact on surrounding marine environments, and as such it fits the scope and potentially can make a good addition to the journal.

AR: Thank you for your review of our manuscript, we highly appreciate your time and work. We have answered all your comments below and revised the manuscript accordingly. As requested by Biogeosciences, the revised manuscript will be uploaded at a later stage after responding to all reviewer comments. There will be a track change version of the manuscript, as well as a clean version including all modifications following your and the 2 other reviewers' suggestions. All the line numbers refer to this clean revised version.

RC: Major critical comments are listed below.

The main conclusions of the authors – that estimation of river OM discharge to the coastal zone cannot be based solely on the data of the main stem far from the deltaic region – is certainly useful for modelers, although not novel and generally agrees with large body of evidences collected for example, by Shirshov's Institute of RAS in Arctic rivers and coastal zones (A.P. Lisitsyn's marginal filters concept suggested more than 30 years ago).

AR: Thank you for this suggestion. The publication mentioned by you (Lisitzin, 1994) will perfectly fit into the introduction, particularly into the sentences stating a high importance of the nearshore zones. Citing the work of Lisitzin will improve our introduction; however, we would like to stress that those published findings do not reduce the novelty of our work, since Lisitzin described the marginal zones of the ocean – the extensive coastal area up to several

hundred km from the delta, where riverine freshwater and saltwater are mixing and major sedimentation processes take place. In our paper, we focused on the Lena Delta itself which is right at the interface between land and sea and thus the "gateway" for riverine discharge supplying the marginal zone of the Laptev sea with organic matter.

L61-63 edited: "*The nearshore coastal zone of the Arctic Ocean (including deltas, estuaries, and coasts) is of great importance as major transformation processes of terrestrial material are expected to take place in these biogeochemically active areas (Tanski et al., 2019; Jong et al., 2020; Sanders et al., 2022)* **as well as its sedimentation (Lisitzin, 1994)**."

We also updated the reference list accordingly:

*Lisitzin, A. P.: A marginal Filter of the Ocean. Okeanologiya, 34 (5), 735-747, 1994, (In Russian)*

RC: The Introduction is well written but it is way too general. Former studies on the POC were not discussed (Semiletov, Kutscher, E. Karlsson). As a result, specific objectives and novelty of this work are unlear; and no new nypothesis is proposed to be tested ( a degree of pOC lost in the deltaic zone or the age and origin of POC could be such hypotheses). In anyway, the authors should clearly position this work with respect to former studied of the Lena River to prove its real novelty.

AR: The importance of these (and other) previous studies investigating POC in the Lena River and its Delta is very much appreciated and respected by us. We were glad to refer multiple times to all studies mentioned (Semiletov et al., 2011;Kutscher et al., 2017; Karlsson et al., 2016) and to highlight them in this manuscript. They were included in the discussion part of the paper already, where they were discussed in great detail (Kutscher et al., 2017) and the data provided by these publications were included into our data plots as the referee points out (Kutscher et al., 2017 and Karlsson et al., 2016).

Nonetheless, we appreciate the suggestion of citing these references in an improved and more focused introduction, which we wrote for the revised manuscript.

RC: The Discussion is very much driven by postulated overwhelming role of phytoplankton in POC, d13C, D14C control in the main stem vs. delta. Without Chl a analysis, or any information on the phytoplankton, such a discussion is not substantiated and suggested explanations have low novelty and probably unwarranted. As a minimal research efforts, the authors could examine their TSM samples by SEM to show the presence of higher amount of diatoms in their deltaic samples vs main stem samples Examination of C/N ratio could aslo help a lot in

distinguishing different sources of POM The discussion and data treatment (Fig 4) also ignore that part of POM may be represented by contemporary vegetation debris (i.e., lignin), especially from larch trees, dominating the Lena catchment

AR: Thank you for your comments and suggestions! As pointed out in the manuscript already, Chl-*a* was unfortunately not measured by us, since these kinds of analyses were not part of the work plan for our expedition. In our manuscript we discuss the potential phytoplankton contribution based on indirect indicators ($\delta^{13}$C of POC and some supporting observation (chapter 4.2.1: weather conditions during the sampling period, nitrate gradient)) and previously published suggestions about the nature of POC depleted in $\delta^{13}$C (Winterfeld et al., 2015; Kutscher et al., 2017; Bröder et al., 2020). The $\delta^{13}$C values measured by us in the Lena Delta were significantly lower than in the river and very low in general. The range of values (between -33.32 and -31.01 ‰) which is lower than $\delta^{13}$C from modern plants contribution (from -28.4 to -27.0 ‰) (Vonk et al., 2017) or for the terrestrial primary production (-27.7 ‰) such as organic and litter layers (Wild et al., 2019) and even lower than $\delta^{13}$C reported in previous studies on phytoplankton contribution in the river waters. Such a low range of values was described for the POC during the algae bloom period (Finlay and Kendall, 2008) and for algae directly (Galimov et al., 2006). Thus, taking into account the above, the explanation of these low values in our opinion could be only a phytoplankton bloom.

As it was already mentioned, $\delta^{13}$C for the contemporary vegetation debris is significantly higher than what we measured in POC. We would like to suggest that its input into the $\delta^{13}$C composition of POC was lower than the one from phytoplankton, thus we cannot see its fingerprint in POC isotopic composition. Nevertheless, we consider that remains and particles of modern C3 plants vegetation are present in the riverine waters, thus we would like bring further clarifications into our manuscript:

L514-522: *"…weapons testing during the 1960s and 1970s; thus, values from organic litter from Russia, Scandinavia, and Alaska were included (Wild et al., 2019).*

***Proposing phytoplankton as one of three main sources of POC does not exclude the input of contemporary vegetation into the riverine and deltaic OM. $\Delta^{14}$C signals from modern vegetation and phytoplankton sources can be assumed to be identical, and similar to the atmosphere (Winterfeld et al., 2015, Wild et al. 2019), while their d$^{13}$C values are likely different. Therefore, phytoplankton was proposed as a modern OM source based on evidence from $\delta^{13}$C values of POC corresponding to algal input (see section 4.2.1) and suggesting sufficiently lower input of plant debits into POC. Modern plants likely contributed as well, but due to their rather constant $\delta^{13}$C signature paired***

*with variable $\Delta^{14}C$ values (plant debris vs roots and litter) they cannot be distinguished from Holocene soils and must be regarded to be a contributor to this endmember."*

We will take additional possible treatments and analyses, which you have kindly suggested in this review into consideration for our further work and future publications.

Specific comments:

RC: L117-120 This might be true; however, di not the former works of Semiletov, Kutscher etc address the transformation of C between Zhigansk/Yakutsk and the delta?

AR: We highly appreciate the study of Kutscher et al. and were glad to refer to this study multiple times in our publication. Nevertheless, the northernmost sampling point in the Kutscher et al. study is Dzhardzhan River (68.7°N, approximately 200 km north from Zhigansk and 600 km south from the Delta).

L118- 123 edited: "*… Zhigansk, located ~800 km upstream from the Lena Delta. **It is also known that a significant fraction of the suspended matter carried by the Lena River is deposited before the Lena reaches Kyusyur, along a narrow part of the Lena main stem called "Lena Pipe" (Semiletov et al., 2011 Fedorova et al., 2016).** As the ArcticGRO sampling location is far from the site where Lena runoff enters the Arctic Ocean and any biogeochemical processes **taking place downstream from Zhigansk and particularly in the delta** are not reflected in the ArcticGRO data, the properties of water and suspended materials sampled at Zhigansk may in fact not be entirely representative of the discharge to the ocean.*"

RC: L176-177 Provide some numbers on the magnitude of Delta14C between "old" and "modern" for non-experienced reader

AR: Changed accordingly:

L186-188 edited: "*Since radiocarbon analysis is commonly used as a method for determining OM age, for discussion of the results, we refer to more $\Delta^{14}C$-depleted samples as "old" **or "ancient"** C from "old" OM sources **(less than -900 ‰ or ~18,500 $^{14}C$ years.)**, and to more $\Delta^{14}C$ enriched samples **(in the range above -50 or ~400 $^{14}C$ years.)**, as "young" or "modern" C.*"

L265-L264 (edited):" *Radiocarbon levels of POC varied within a wide range between -243 and -88 ‰ **(translating to approximately 2236 and 740 $^{14}C$ years mean age, respectively)**"*

RC: L185-187 Neglecting the beginning of spring flood may underestimate sizable amount of riverine C, transported to the delta (which is not the case for the winter time). Justification ere is needed.

AR: The authors' team are fully aware of the crucial importance of the spring flood period for the entire annual C balance of the Lena River discharge. Nevertheless, the aim of this study was not to estimate the annual OM discharge, but to compare POC discharge and its sources in the Lena main stem and delta based on samples collected in summer 2019, during a time not affected by the spring flood conditions.

RC: L197-198 Former studies already shown this; why additional efforts are needed?

AR: Since stratification is one of the crucial parameters for estuaries in terms of their functioning (Geyer and Ralston, 2012) we found it essential to investigate if this parameter was reflected in the Lena Delta during sampling. Thus, we would like to keep mentioned information about our results (reference to the Fuchs et al., (2022), which is based on the same samples as this study) as a clarification of deltaic conditions. We further stress this here, as the lack of stratification is distinct from other Arctic rivers (e.g., Mackenzie, (Hilton et al., 2015)).

RC: L203-205 Unclear. If there is no difference in deltaic region (L197-198), why there should be any in the river main stem? More likely explanation is due to seasonal variations in C concentrations in the Arctic GRO dataset.

AR: We found this clarification essential, since potentially river water masses may be stratified as well, which is a reason for the ArcticGRO group to organize depth-integrated sampling. We added this explanation to the manuscript briefly:

L217-226: "*We did not collect samples from different water depths along the river transect from Yakutsk to Stolb but instead were only able to sample surface waters. In contrast to our surface water samples, ArcticGRO samples are depth-integrated, **since potentially river water masses may be stratified (e.g. Mackenzie: Hilton et al., 2015)**. This difference in sampling might explain some of the differences between our observations and those made by ArcticGRO (see sections 3.3.1 & 4.2.2**)."*

Reference list updates:

*Hilton, R., Galy, V., Gaillardet, J., Dellinger, M., Bryant, C., O'Regan, M., Gröcke, D.R., Coxall, H., Bouchez, J. & Calmels, D. Erosion of organic carbon in the Arctic as a geological carbon dioxide sink. Nature 524, 84–87, https://doi.org/10.1038/nature14653, 2015*

RC: Fig. 2 is well presented. However, the data of former researchers, obtained at these transects (at least, the Yakutsk – Kusur one) should be also presented

AR: We appreciate this suggestion! We had considered including published data in Fig. 2 but finally decided against doing so. This is based on three reasons: 1) results from former studies are presented in Fig 3 and 4 already; 2) Fig 2 was dedicated to our results measured for this particular sampling campaign in summer 2019. We would like to keep the focus on our sampled sites; 3) Additional datasets would make the figure too crowded and would distract from our main findings.

RC: Section 4.1.1 can be strongly shortened; the novelty of these findings is low. Summarize in one paragraph. Some relevant information can be shifted to the caption of Fig. S1.

AR: Thank you, following your advice we shortened section 4.1.1 (L294-317):

*"The Lena River is characterized by a nival hydrograph regime with a distinct flood event taking place in the beginning of summer during the snowmelt and ice breakup period (May-June) and a very low water flow in winter (Yang et al., 2002). Discharge has a strong effect on the amount of solids and OM released by a river (Magritsky et al., 2018). The peak of annual POC concentrations in the Lena River (>3.6 mg $L^{-1}$, McClelland et al., 2016) and TSM concentrations (>150 mg $L^{-1}$, ArcticGRO) occur right after the flooding following ice breakup in late May–early June.*

*The peak water yield in 2019 took place on 2 June and reached 83,000 $m^3$ $s^{-1}$, then it decreased and varied in the range of 49,200–45,999 $m^3$ $s^{-1}$ during the time interval when the main stem transect was sampled. During the sampling in the Delta (2019/08/07–2019/08/09) the discharge was 19,600–19,000 $m^3$ $s^{-1}$, which was less than half the discharge during main stem sampling.*

*We analysed all ArcticGRO data on TSM and POC for the Lena River to demonstrate that TSM and POC concentrations were correlated with discharge (Figure S1). However, when considering a discharge ranging between 15,000 and 50,000 $m^3$ $s^{-1}$, (the typical range of discharge values in summer (including 2019) and in September), there is no significant relationship between discharge and TSM and POC concentrations (Figure S2). Thus, the strong relationship between discharge and TSM/POC appears to be driven by the large difference between maxima in all parameters (observed during the spring flood) and their minima (found during low flow in winter).*

*Average surface water TSM and POC concentrations in the Lena River in 2019 agree with reported average TSM and POC concentrations observed by ArcticGRO during periods with discharge within the mentioned range (TSM in this study: 21.29 mg $L^{-1}$ as compared with 22.66 mg $L^{-1}$ for ArcticGRO, and POC 0.77 mg $L^{-1}$ and 0.79 mg $L^{-1}$, respectively). TSM and POC concentrations in the Lena Delta in summer 2019 were 2 and 1.5 times lower, respectively, than values reported by ArcticGRO for a comparable time of year and under similar discharge conditions (TSM: 9.3 ±5.2 mg $L^{-1}$, POC: 0.41 ±0.10 mg $L^{-1}$) (Figure 2a & b). On the other hand, our deltaic POC concentrations are similar to previously published POC data for the Lena Delta (Winterfeld et al., 2015) (Figure 3). This shows that the difference we observe between river and delta is a persistent feature that is not biased by sampling time or depth but is mostly caused by other factors such as, e.g., flow and velocity."*

RC: L314-320 This is site description; re-arrange

AR: Rearranged accordingly, we also have edited this paragraph accordingly to the advise of the third reviewer.

L97-105: *"The Lena River watershed was subdivided into the Upper and the Lower Lena, which contribute differently to the TSM and water discharge into the Lena River and are characterised by distinct morphologies. **Here, we define the Upper and Lower Lena River by the area of subcatchments of the Lena River (https://www.hydrosheds.org/products/hydrobasins). The separation between the Upper and Lower Lena was made approximately 150 km downstream from Yakutsk (Figure 1a). The Upper Lena includes the southern limits of the river and the catchment upstream of the Aldan junction., Its watershed covers an extensive area between Lake Baikal and Yakutsk and includes dozens of tributaries including creeks and small rivers. The Lower Lena consists of the catchment area downstream of the Aldan junction excluding the catchments of Aldan and Vilyuy (Figure 1a). It flows from downstream of Yakutsk into the Laptev Sea and receives waters from catchments including the Verkchoyansk Range.**"*

RC: L353 Present the numbers of velocities in thee regions

AR: Changed accordingly: L352-354: *"... to settle rapidly. Thus, decreasing velocity in the Lower Lena (**from 2.5 m $s^{-1}$ to 0.8 m $s^{-1}$ in May (Kääb et al., 2013)**) and the Lena Delta itself (**from 1.3 to 0.9 m $s^{-1}$ in August (Nigamatzyanova et al., 2015**) allows further sedimentation of TSM and old C, resulting in a decrease of its concentration."*

Added to the Reference list: *Nigamatzyanova, G.R., Frolova, L.A., Chetverova, A.A., Fedorova, I.V.: Hydrobiological investigation of channels in the mouth region of the Lena River, Uchenye Zapiski Kazanskogo Universiteta. Seriya Estestvennye Nauki, vol. 157, no. 4, pp. 96–108, 2015, (In Russian)*

RC: L420 There should be some data for the man stem

AR: We agree with the reviewer that such data would be very valuable, but have so far not been able to find published $\delta^{13}C$ values of DIC in the Lena Delta. We would be most grateful if the reviewer would point us to references that we might have overlooked.

Nevertheless, we have changed the focus of this sentence from literature values to our results to avoid any possible misunderstanding. Thus, we state that measurement of $\delta^{13}C$ of DIC was not available, and not that it was never done before.

L420 edited: "*In the Lena Delta the $\delta^{13}C$ of DIC has not been **measured**, but the low $\delta^{13}C$ of POC…*"

RC: L439 d13C of POC?

AR: Thanks, we clarified this: L439 edited: "*$\delta^{13}C$ of **POC** from ArcticGRO was lower than the $\delta^{13}C$ of POC we measured in the main stem, which may indicate more phytoplankton…*"

RC: L558-563 The novelty of the present study seems to be low

AR: We disagree respectfully with this statement. With our research we provided a direct and detailed comparison of POC in the Lena River (measured along the transect from Yakutsk to the Lena Delta) and in the Lena Delta (along the entire Sardakhskaya branch). Our findings are based on C isotopic composition analyses of POC which require some state of the art facilities which were unfortunately not available for the excellent, deep and extensive studies on the region from the past (30 years ago and earlier). Modern studies, where C isotopic composition was analysed usually focus on the wide coastal zone of Laptev sea (for example as L558 Karlsson et al 2016 and Semiletov et al., 2011) or the river borne OM (Wild et al., 2019, etc, Semiletov et al., 2011). In both cases, the studies do not consider the Lena Delta and its role in the balance and composition of OM discharge. With this manuscript, we would like to fill in this gap and to highlight the high importance of the Delta in terms of the Lena River OM discharge to the ocean. Thus, with our study we built a bridge between two major approaches of investigations in this area: study of the riverine OM or Laptev Sea shelf investigations. Finally, we see this manuscript as a crucial piece into the puzzle of understanding of the Lena River – Laptev Sea interaction. Especially in the context of ongoing

climate change and permafrost degradation as it was demonstrated in L558-563, where we showed the evidence of an additional contribution of Yedoma to OM discharged by the Lena River to the Laptev Sea, which takes place particularly in the Delta. This finding was not demonstrated and published before.

To make this clearer we edited L79-80: "*In this study, we aim **to bridge this gap and to characterise…***"

The team of authors would like to thank AR for the work, time, editing and contribution to our manuscript and wish all the best!

References used in this response:

Bröder, L., Davydova, A., Davydov, S., Zimov, N., and Haghipour, N.: Particulate Organic Matter Dynamics in a Permafrost Headwater Stream and the Kolyma River Mainstem Journal of Geophysical Research : Biogeosciences. 1–16, https://doi.org/10.1029/2019JG005511, 2020

Geyer, W. R., and D. K. Ralston. 2012. The Dynamics of Strongly Stratified Estuaries. Treatise on Estuarine and Coastal Science. Vol. 2. Elsevier Inc. https://doi.org/10.1016/B978-0-12-374711-2.00206-0

Finlay, J. C., and Kendall, C.: Stable Isotope Tracing of Temporal and Spatial Variability in Organic Matter Sources to Freshwater Ecosystems. In Stable Isotopes in Ecology and Environmental Science: Second Edition (pp. 283-333). Blackwell Publishing, https://doi.org/10.1002/9780470691854.ch10, 2008

Fuchs, Matthias, Juri Palmtag, Bennet Juhls, Paul Overduin, Guido Grosse, Ahmed Abdelwahab, Michael Bedington, et al. 2022. "High-Resolution Bathymetry Models for the Lena Delta and Kolyma Gulf Coastal Zones." Earth System Science Data Discussions, 1–30.

Hilton, R., Galy, V., Gaillardet, J. et al. Erosion of organic carbon in the Arctic as a geological carbon dioxide sink. Nature 524, 84–87. https://doi.org/10.1038/nature14653, 2015

Karlsson, E., Gelting, J., Tesi, T., van Dongen, B., Andersson, A., Semiletov, I., Charkin, A., Dudarev, O., and Gustafsson, Ö.: Different sources and degradation state of dissolved, particulate, and sedimentary organic matter along the Eurasian Arctic coastal margin, Global Biogeochem. Cycles, 30, 898– 919, https://doi:10.1002/2015GB005307, 2016

Kutscher, L., Mörth, C.-M., Porcelli, D., Hirst, C., Maximov, T. C., Petrov, R. E., and Andersson, P. S.: Spatial variation in concentration and sources of organic carbon in the Lena River, Siberia, J. Geophys. Res. Biogeosci., 122, 1999– 2016, https://doi:10.1002/2017JG003858, 2017

Lisitzin, A. P.: A mariginal Filter of the Ocean. Okeanologiya, 34 (5), 735-747, 1994, (In Russian)

Nigamatzyanova, G.R., Frolova, L.A., Chetverova, A.A., Fedorova, I.V.: Hydrobiological investigation of channels in the mouth region of the Lena River, Uchenye Zapiski Kazanskogo Universiteta. Seriya Estestvennye Nauki, vol. 157, no. 4, pp. 96–108, 2015, (In Russian)

Vonk, J. E., Tesi, T., Bröder, L., Holmstrand, H., Hugelius, G., Andersson, A., Dudarev, O., Semiletov, I., and Gustafsson, Ö.: Distinguishing between old and modern permafrost sources in the northeast Siberian land–shelf system with compound-specific δ2H analysis, The Cryosphere, 11, 1879–1895, https://doi.org/10.5194/tc-11-1879-2017, 2017

Winterfeld, M., Laepple, T., and Mollenhauer, G.: Characterization of particulate organic matter in the Lena River delta and adjacent nearshore zone, NE Siberia – Part I: Radiocarbon inventories, Biogeosciences, 12, 3769–3788, https://doi.org/10.5194/bg-12-3769-2015, 2015

---

## Author Comment (AC2)

**Discussion on the 2nd Anonymous Referee review**

Referee comment on "Particulate organic matter in the Lena River and its Delta: From the permafrost catchment to the Arctic Ocean" by Olga Ogneva et al., Biogeosciences Discuss., https://doi.org/10.5194/bg-2022-183-RC2, 2022
* * *
Comment types: Authors' Response: "AR", Referee Comment: "RC"

Comment colors: Authors Response: "blue", Referee Comment: "black"

Comment fonts: When it was possible, we highlighted changed text by the **bold font**, the text from the manuscript copied to this review was taped *cursive*

RC: This paper reports majorly the measurement of TSM, POC, $\delta^{13}C$ and $^{14}C$ in the Lima River during 2019, a year that represented "lower-than-average" TSM exports and showcased a strong positive influence on phytoplankton growth. The paper highlights the importance of deltaic processes. Findings are potentially important because they inform how climate change may influence Arctic carbon fluxes to the ocean. However, the paper has a few areas that require improvement. The authors fail to provide a discharge time series for the year of data collection and ArcticGRO sampling period to put their findings in context. The paper can be improved if the following changes are made:

AR: Thank you for your review of our manuscript, we highly appreciate your time and work. We have answered all your comments below and revised the manuscript accordingly. As requested by Biogeosciences, the revised manuscript will be uploaded at a later stage after responding to all reviewer comments. There will be a track change version of the manuscript, as well as a clean version including all modifications following your and the 2 other reviewers' suggestions. All the line numbers refer to this clean revised version.

RC: In the introduction section, explain the importance of Lena River and why it is important to study it in 3 sub-sections (as mentioned in lines 180-183). Further, add statements about the research gap and focus of the current study.

AR: Thank you for suggestions! We have changed our introduction to highlight the research gap and the importance of the Lena River and its Delta, which also clarifies the subdivision of the data set/study area.

L73-78: "*Thus, for example, sampling from the Lena River,* **which transports the largest amount of particulate organic C (POC) of all Arctic rivers to the Arctic Ocean (McClelland et al., 2016) and has one of the world's biggest deltas,** *took place ~ 800 km upstream from the Lena Delta at the town of Zhigansk. This long distance of the sampling site from the areas, where the river enters the Arctic Ocean, and the*

*deficit of information about the delta and the potential biogeochemical processes taking place there (OM transformation/sedimentation/enrichment) which may lead to a distortion or a lack of information about the final state of OM reaching the Arctic Ocean."*

*L79-82: "In this study, we aim **to bridge this gap and to** characterise POC along the Lena River over a transect from upper reaches of the Lena River near Yakutsk (approximately 1640 km from the coast) north to the Lena Delta in order to decipher the distribution, main sources, and transformation of particulate organic matter (POM) on its way from the permafrost catchment to the Arctic Ocean."*

RC: Figure 1: Include the information on the sample number in the caption. Also, try to showcase three divisions of sample groups for easy understanding

AR: The Figure 1 and the caption for Figure 1 were modified as suggested:

[Figure]

*"Figure 1. The Lena River catchment and its Delta with Yedoma distribution (Strauss et al., 2021b) in the catchment. a) Sampling locations along the Lena River main stem (n of samples = 9) and the Lena River catchment area; b) Sampling locations at Stolb Island (n of samples = 3) and in the Lena Delta along the Sardakhskaya branch (n of samples = 20)."*

RC: Figure 2: Edit and add mean values of your results and ArcticGRO which you are discussing in Sections 3.2.2 to 3.2.4

AR: Figure 2 and its caption were modified as:

[Figure]

*"Figure 2. Distribution of studied parameters along the transect in the Lena main stem, the Lena Delta, and for the ArcticGRO dataset (mean ±stdev); values on each panel represent the average (±stdev) for every sampling group: a. TSM mg L-1; b. POC, mg L-1; c) OCTSM, wt%; d) Δ14C of POC, ‰; e) δ13C of POC, ‰."*

RC: Show the river discharge time series data relative to ArcticGRO. It is necessary to fully interpret these results.

AR: *We added this information to the supplements and refer to it within the text:*

L36-307: *"…relationship between discharge and TSM and POC concentrations* **(Figure S2)***…"*

[Figure]

*Figure S1: Time series of the Lena River discharge provided by ArcticGRO for 2003-2019 and TSM, POC concentration fluctuation.*

RC: Lines 301-302 mention that 2019 was a year of lower-than-average TSM export. Discuss the variation in TSM and POC on a large timescale and present a plot of temporal variation for better clarity.

AR: This chapter (4.1.1) was shortened a lot according to suggestions of the 1st anonymous review. Thus, the revised manuscript does not contain the discussion on the POC and TSM fluctuation any more. To get your suggestion integrated we would like to note that in the new version of the chapter we referred to the Supplement, which includes the requested additional information about TSM, POC, and annual fluctuation (Figure S1, S2).

RC: Lines 309-310: Provide a figure or table comparing the POC variation with the published data

AR: Thank you. We would like to point the reviewer to the sentence cited below from our original manuscript, where we refer to a figure comparing the published data with our new results, which he/she perhaps accidentally overlooked.

L315-316: *"On the other hand, our deltaic POC concentrations are similar to previously published POC data for the Lena Delta (Winterfeld et al., 2015)* **(Figure 3)***."*

RC: Also utilize discharge data to calculate the flux of TSM, and POC and compare it with previous reports.

AR: That would be a very interesting topic to highlight! Nevertheless, the calculation of OM fluxes are beyond the scope of this manuscript. Further, we would like to mention that intensive hydrological measurements at our sampling sites would be needed to provide such calculations. Any attempts to estimate the discharge on our database which was not designed for such calculations would lead to misleading numbers. We therefore

would like to refrain from the calculation of TSM and POC fluxes.

RC: Section 4.1.1 and 4.1.2: Again, it would be helpful to see the discharge time series for the ArcticGRO period of sampling vs other years such as 2019. These variations in POC% are hard to interpret without seeing the discharge time series. Further, it is often helpful to calculate the ratio of the coefficient of variation (CV) of your parameter (e.g., POC%) to the CV for Q; CVc/CVq to determine how much discharge is affecting the variation.
AR: The time series plot for the ArcticGRO was added to the supplement (Figure S1). Concerning your question on the additional statistical parameters for our data, such a detailed analysis of discharge was not the aim of our work. Moreover, we have changed section 4.1.1. by shortening it according to the first anonymous review. In detail, we removed potentially misleading sentences about POC variation.

RC: Minor comment: Check the mention of the figure numbers in the text. There is no figure 3c
AR: Thank you very much for mentioning this! We changed this to fig 3a
L381: "*In contrast to main stem POC, deltaic POC was more depleted in $\delta^{13}C$ (mean -32.5 ±0.7 ‰) (Figure 2e; **3a**)*".
L413: "*…weeks (11 July – 25 July 2017), suggesting that an algal bloom developed between the two sampling dates (Figure **3a**).*"

RC: Lines 420-421: Provide a reason for not analyzing δ13C of DIC. Additionally, you need to provide reasons why you considered that low δ13C of POC suggests a 13C depleted DIC pool with more references.
AR: Unfortunately, due to the complicated logistics in the region of sampling, difficulties occurred during the transit of samples from the Lena Delta to the Bremerhaven. We were therefore unable to analyse the samples we initially took for $\delta^{13}C$ of DIC. The citation was put into the right place (Brunet et al., 2005) see the comment below.

RC: Line 420: $\delta^{13}C$ of DIC was found to be negatively correlated with DOC concentration. Is this the observation of Brunet (2005)? If yes, please rewrite the sentence with the proper citation.
Thank you, we changed this accordingly.
L420-423: "*suggests a $^{13}C$-depleted DIC pool. Low $\delta^{13}C$ in DIC can be caused by several processes as shown for riverine DIC in the rivers of Patagonia: degradation of dissolved OC (DOC) containing soil organic C will result in low $\delta^{13}C$ of DIC, and $\delta^{13}C$ of DIC was found to be negatively correlated with DOC concentration **(Brunet et al., 2005)**.*"

RC: General comment. Do your data suggest any influence of lakes on your TSM and POC concentration?        How        do        you        rule        them        out?

AR: This question was not taken into account during our research, despite the fact that we find it very interesting! We suggest that the input of lakes despite their very particular role in the C turnover system and their effect on qualitative and quantitative characteristics of Lena POM would not be significant due to the Lena extensiveness compared to the lakes as it was shown for DOC (Stolpman et al., 2022).

The team of authors would like to thank AR for their work, time, editing and contribution to our manuscript and we wish them all the best!

References used in this response:

Stolpmann, L., Mollenhauer, G., Morgenstern, A., Hammes, J.S., Boike, J., Overduin, P.P., and Grosse, G.: Origin and Pathways of Dissolved Organic Carbon in a Small Catchment in the Lena River Delta, Front. Earth Sci. 9:759085. doi: 10.3389/feart.2021.759085, 2022

---

## Author Comment (AC3)

**Discussion on the 3rd Anonymous Referee review**

Referee comment on "Particulate organic matter in the Lena River and its Delta: From the permafrost catchment to the Arctic Ocean" by Olga Ogneva et al., Biogeosciences Discuss., https://doi.org/10.5194/bg-2022-183-RC3, 2022
* * *
Comment types: Authors' Response: "AR", Referee Comment: "RC"

Comment colors: Authors Response: "blue", Referee Comment: "black"

Comment fonts: When it was possible, we highlighted changed text by the **bold font**, the text from the manuscript copied to this review was taped *cursive*

RC: The manuscript presents and discusses the data on quantity and quality of suspended matter, including its organic fraction, in the Lena River main channel in its middle and lower sections, and in the deltaic section. The manuscript has logical structure, is clearly written, presents novel data and adds to the discussion on the fate of particulate organic matter at the interface between the river and the sea. Two major points arise from the endmember modeling of the POC isotopic signatures : an important contribution from phytoplancton in both riverine and deltaic sections, and a noticeable input from the Ice Complex deposits in the Lena Delta region. The manuscript text needs to be more focused on these two findings, and if the authors think I have omitted any other important aspects, these aspects also need to be clearly framed and put forward. Notably, the 'Discussion' section of the manuscript needs to be centered around these major findings ; it is vague and uninspiring in its present form. Certain sections of this Discussion section, i.e. subsections 4.1.1 and 4.1.2, are quite detached from hydrologic reality. The comparison with ArcticGRO data makes a large part of discussion, while the discussion is usually self-sufficient and mostly relies on the newly presented data. I would suggest a separate section named 'Comparison with previously published datasets' for this discussion rather than spreading them across different sections.

Overall, I recommend a moderate revision of this manuscript with re-review. The revision might affect, first and foremost, the Discusson section, in what is related to regional hydrology, and better framing the major conclusions. By-line comments are available in the attached pdf.

Please also note the supplement to this comment: https://bg.copernicus.org/preprints/bg-2022-183/bg-2022-183-RC3-supplement.pdf

AR: Thank you for your review of our manuscript, we highly appreciate your time and work. We have answered all your comments (including those from the supplemental pdf, which now are transferred into this document with the specific line number) below and revised the manuscript accordingly. As requested by Biogeosciences, the revised manuscript will be uploaded at a later stage after responding to all reviewer comments. There will be a track

change version of the manuscript, as well as a clean version including all modifications following your and the 2 other reviewers' suggestions. All the line numbers refer to this clean revised version.

Thank you for your comments on our discussion. We respectfully disagree with the need to change the structure of the discussion part for a number of reasons we would like to explain. Our aim was to characterize, as fully as possible, the POM in the Lena River and its Delta, as well as to compare them. We are convinced that the current sequence of discourse and observation is the most straightforward way for our aims to provide the entire picture of research.

We agree with you on the two main findings, which are now more clearly described in the abstract and conclusion parts. Specific comments and edits were revised accordingly and are listed below.

L24-26: "*Dual-carbon ($\Delta^{14}C$ and $\delta^{13}C$) isotope mixing model analyses* **indicated a significant phytoplankton contribution to deltaic POC (~68 ±6 %) and suggested an additional input of permafrost-derived OM into deltaic waters (~18 ±4 % of deltaic POC originates from Pleistocene deposits vs ~5 ±4% in the river main stem)***.

RC: L19-20: In the Abstract, you mostly present the results concerning the river-sea interface, and as early as in the Abstract the question arises what results were obtained from the transect itself. The Abstratc is uninformative in this regard.

AR: Thank you. We rearranged the abstract and edited L19-20 as suggested: *"Here, we studied **particulate organic carbon (POC) dynamics in the Lena Delta and compared it with POC dynamic in the Lena River main stem** along a ~1600 km transect long from Yakutsk downstream to the delta. We measured **POC**…"*

RC: L20: "Disembogue" is rather used as a verb, i.e. 'to disembogue', than as a noun according to Collins Dictionary.

AR: We changed this sentence and removed the word "disembogue"(L20).

RC:L28: I wonder if there is any difference between 'the Pleistocene deposits' in the delta area and 'the Yedoma' in the Lena R. catchment?  This phrase creates an enormous confusion, comparable to 'Late Pleistocene Yedoma' from certain Canadian authors. It is either this or that, or Ice Complex deposits

AR: Thank you, changed accordingly at L28: *"**Yedoma** deposits in deltaic waters was almost twice as large as POC of Yedoma origin…"*

RC: L45: Lotsar – i

AR: Thank you very much for your attentiveness! We changed this as suggested to "*Lotsari*" (L45)

RC: L49: Correct me if I'm wrong but delayed active layer freeze-up was not in the scope of

(Fuchs et al., 2020) paper on Sobo-Sise cliff, but rather fluvial thermal erosion and abrasion were.

AR: Your are right, so we removed this misplaced reference (L47-48):

"*In addition, the delayed active layer freeze-up increases winter river runoff (Walvoord & Kurylyk, 2016; Lamontagne-Hallé et al., 2018; Wang et al., 2021a).*"

RC: L49 "enchances/Enchanced" – repetitive

AR: Thank you! See comment above: the first sentence was modified, thus the word "Enchanced" at the L49 is not repetitive anymore.

RC: L86-87: No data from official sources are available for the Lena R. at Kyusyur after 2013.

RC: L114-115: This is highly dubious, since Wang et al. 2021b paper and conclusions can not be traced to verifiable data; also, the mean annual Lena River discharge/runoff for years 2014-2020 can not be accessed because of incomplete officially published Roshydromet data from Kyusyur gauging station. The most recent data-supported trend estimate is 15.6% (1936-2013) from Tananaev et al., 2016 GRL.

RC: L188-189: Official data are incomplete for the Kyusyur g/s for 2019, therefore I wonder if the data are correct, and what is the actual source of these data; the following material suggests that no, not entirely, e.g. see below data on peak discharge.

AR: We would like to combine these three comments associated with the same issue and to provide one detailed answer for all of them. The discharge data for the Lena River collected by Roshydromet as well as for several other large Arctic rivers were published on the base of ArcticGRO project (Shiklomanov et al., 2021). These data could be accessed and used from this source: https://arcticgreatrivers.org/data/. At the moment of writing this response (November 2022) it is available for Kyusyur station until 29 May 2022. Moreover, there are a number of studies that use Lena River discharge data from ArcticGRO after the year 2013. We invite you to browse a good collection of them here: https://arcticgreatrivers.org/publications/ .

To make this clear we adjusted L199-202: "T*he discharge data are provided by the Russian Federal Service for Hydrometeorology and Environmental Monitoring (Roshydromet,* **published by Shiklomanov et al., 2021b**) *for the Lena River at Kyusyur (70.68°N, 127.39°E, see Figure 1a).*"

We also updated the reference list accordingly:

Shiklomanov, A., Déry, S., Tretiakov, M., Yang, D., Magritsky, D., Georgiadi A., Tang W.: River Freshwater Flux to the Arctic Ocean. In: Yang, D., Kane, D.L. (eds) Arctic Hydrology, Permafrost and Ecosystems. Springer, Cham., https://doi.org/10.1007/978-3-030-50930-9_24, 2021**a**

**Shiklomanov, A.I., R.M. Holmes, J.W. McClelland, S.E. Tank, and R.G.M. Spencer. Arctic Great Rivers Observatory. Discharge Dataset, Version YYYYMMDD.**

**https://www.arcticrivers.org/data, 2021b**

RC: Figure 1: Vilyuy

AR: We changed Figure 1 accordingly:

[Figure]

The name of the river was changed within the entire manuscript text accordingly as well: (see L104, 281, 324, 328, 341)

RC: L129-130: Is it correct that the sample volume for TSM analysis was less than 1L ? Though these samples evidently can not be retaken, it is assumed that a sample of at least 1L is needed to assure correct TSM measurement (in mg L-1). Sample freezing before filtration is also a questionable practice, since we have numerous times observed organic matter coagulation in (presumably) ferro-organic colloidal complexes upon freezing which would normally not be included in TSM if not frozen.

AR: In 2012, the "optimal filtration volume" for TSM analyses was established by Neukermans et al., (2012). They stated that an optimal sample volume is dependent on the amount of particles retained on the filter and must contain enough TSM to be sufficiently and precisely measured. Therefore, if enough particles are present in a volume less than 1 L, this is the optimal sample volume. Neukermans has also shown that it is possible to use a lower or higher volume of water as well and TSM concentration stays stable.

Additionally, it was further shown by Röttgers et al (2014) that the volume of the sample does not have a significant impact on TSM concentration variation (water samples from Elbe

estuary of volumes from 100 ml to several Liters were studied). Thus, it is appropriate to use an aliquot of less than 1 L for TSM measurement, when this water was taken from the river/estuary. The protocols you are referring to are often originating from the marine community sampling in clear oceanic waters with very low matter concentrations, which is definitively not applicable for Lena River and Delta water.

[Figure]

**Fig. 8.** Boxplot of [SPM] obtained through filtration of different volumes of sampled seawater at six different stations.

Figure 8 from Neukermans et al (2014).

Concerning your question on freezing the samples: In our work we avoided sample freezing when it was possible and filtrations could be done immediately. However, this could not always be realized. Thus we used immediate sample freezing for the water preservation. It was shown by Nachimuthu et al., (2020) that freezing of unfiltered water samples leads to variable DOC results, but is an appropriate treatment in case of total organic carbon determination.

RC: L186: TSM, SSC, SPM there are too many acronyms here, please reduce to one
Thank you, changed accordingly. L197: "..*spring – early summer ice breakup (maximum water and TSM discharge)*"

RC: L195: 'Discussion' if has important implications, otherwise in methods description. Also important whether a cross-section profiles were done at
AR: Since it is known that water profile of the river and estuary may be stratified (for example Geyer and Ralston, 2012), we suggest this result (the absence of stratification in the Lena Delta) is very important to be mentioned in the "Results" part of the paper. Cross-section profiles were not done. We sampled along a transect. This information is described in the methodology part (2.2. Sampling).

RC: L222-223 Georgiadi et al. 2019 ?
AR: We decided against referring to the mentioned work in that part of our paper because L

222-223 belongs to the result part of the paper. Here we would like to keep describing our own results and comparing them with ArcGRO dataset. Additionally, the manuscript mentioned does not include information regarding the TSM.

RC: L235: Georgiadi et al. 2019 ?

AR: Unfortunately, despite the huge interest of our group in the great work of A. Georgiadi , N. Tananaev and L. Dukhova and even an attempt to combine our datasets, we cannot directly compare our POC data with the data provided by this publication due to methodological differences. POC reported by Georgiadi at al. was determined as the loss of ignition using the theoretical calculation of C as 45% of OM lost during ignition and without pre-acidifying the filter, whereas we determined POC directly after getting rid of inorganic C by acidifying the filter.

RC: L275: The dataset is not a research subject but the TSM and POC are. So, not 'the dataset displayed', but 'the concentrations displayed'.

AR: Changed accordingly with editing (L288): "*Our TSM and POC concentrations measured in 2019 displayed generally higher and more variable values…*"

RC: L276: This preliminary suggests that further discussion will evolve around explanations why the TSM/POC concentrations are more variable in the Lena R. than in the delta. But is this accurate?

AR: This paragraph is an introduction for the detailed part of the discussion on the TSM/POC concentrations, which follows this paragraph as separate chapters (4.1.1 and 4.1.2).
A reference to Figure 2, which demonstrates that OC content in TSM is higher in the Lena Delta than in the Lena main stem was added.
L289: "*…while the OC content of TSM was higher in the Lena Delta* **(Figure 2c)**."

RC: L285: 111,000 m3 s-1 according to official Roshydromet data; also provide values for Tabaga g/s since the transect starts there. A hydrogram will be useful to better relate the survey to actual hydrological conditions.

AR: Discharge data for the Lena River collected by Roshydromet as well as for several other large Arctic rivers were published on the base of ArcticGRO project, which we were happy to find and use here: https://arcticgreatrivers.org/data/.

RC: L301-302: No, unless you present data from spring freshet period, which comprises about 80 to 90% of suspended sediment runoff.

AR: Thank you. This sentence was removed following also the suggestion from the first anonymous referee.

RC: L315-316: This is incorrect. The Upper Lena R. is upstream Vitim, the Middle Lena is from Vitim to the Vilyuy R. mouth, and Lower Lena R. downstream the Viliyuy R.  The discussion

that follows should be better aligned with hydrological reality of the catchment.

AR: Here, we define the Lower and Upper Lena River by the area of subcatchments of the Lena River. We did not subdivide Middle Lena as well as for example Kutscher et al., 2017 and Liu et al., 2005.

To define these subcatchments we used the most recent data that is available: https://www.hydrosheds.org/products/hydrobasins. As a result, while the Upper Lena River includes the catchment upstream of the Aldan junction, the lower Lena River consists of the catchment area downstream of the Aldan junction excluding the catchments of Aldan and Vilyuy.

To avoid any misunderstanding regarding the definition of the Upper and Lower Lena we added this essential clarification into our manuscript and to edit our previous description:

L99-105: *"The Lena River watershed was subdivided into the Upper and the Lower Lena, which contribute differently to the TSM and water discharge into the Lena River and are characterised by distinct morphologies.* **Here, we define the Upper and Lower Lena River by the area of subcatchments of the Lena River (https://www.hydrosheds.org/products/hydrobasins). The separation between the Upper and Lower Lena was made approximately 150 km downstream from Yakutsk (Figure 1a). The Upper Lena includes the southern limits of the river and the catchment upstream of the Aldan junction, its watershed covers an extensive area between Lake Baikal and Yakutsk and includes dozens of tributaries including creeks and small rivers. The Lower Lena consists of the catchment area downstream of the Aldan junction excluding the catchments of Aldan and Vilyuy (Figure 1a). It flows from downstream of Yakutsk into the Laptev Sea and receives waters from catchments including the Verkchoyansk Range***.*"

RC: L318-320: No
AR: Please take a look at our response to the previous comment. There, we explained our approach regarding the Lena River watershed subdivision.

RC: L333-334: The Lena River is not fed from the Central Siberian Plateau, except minor tributaries of the Vilyuy R. which is not, in any case, make part of the Upper lena River basin as declared here. The Aldan R. catchment never was 'relatively flat', and is flanked by the mountains all along the right-bank side, and the Aldan River itself originates from the mountainous region.
AR: To avoid misunderstandings we removed this sentence from the manuscript.

RC: L342: Vilyuy
AR: Yes, following your advice we replaced Vilyuy accordingly here and the rest of the manuscript: L104, 281, 324, 328, 341.

RC: L346: Kyusyur
AR: replaced accordingly and multiply along the entire manuscript (L126, 200, 324, 344, 345,

346, 400, 561) and at Figure 1 and 2.

RC: L351: Transporting ed

AR: Changed accordingly (L349).

RC: L347-348: If it is known, please reference.

AR: Changed accordingly in L345-347: "*It is also known that the majority of TSM brought by the Lena River from the water catchment is deposited before the Lena reaches the stream north of Kyusyur, which is known as the "Lenskaya Truba" (which means "Lena pipe") (Antonov, 1960).*"

Reference list update:

Antonov, V.S.: The Lena River Delta.- Works of oceanographic committee of the Acad. Sci. USSR Vol. VI: 25-34, 1960 (in Russian)

RC: L442: 9 nine

AR: Edited accordingly (L441)

RC: L475-477: Any satellite imagery available for the survey dates to track river turbidity at sampling locations, and/or ChlA quantity ?

AR: Tracking river turbidity and/or chlorophyll-a in Arctic rivers using remote sensing is not a straightforward task. Thus, available products often lead to a misinterpretation of real conditions.

For instance, backscattering of phytoplankton in surface waters can result in stronger backscattering that could be misinterpreted as higher sediment particle load.

The Lena River as all Arctic rivers and Arctic coastal waters are optically complex waters. Optically, the water is dominated by the absorption of organic matter and existing algorithms currently fail to retrieve chl-a and sediment/turbidity load (IOCCG, 2015).

RC: L553-554: No need to explain what you show on the Figure - there is a figure caption for this. Figures are illustrative to your ideas - rather present ideas than figures.

AR: Changed accordingly (L559-562): "*The estimated contributions of different OM sources to the actual average POC concentration measured for the Lena main stem and delta (Figure 5) showed that POC derived from Holocene soils decreased from 0.44 to 0.05 mg L$^{-1}$ due to sedimentation, which takes place in the Lower Lena, particularly downstream from Kyusyur station, and in the delta itself as explained in section 4.1.2.*"

RC: Figure 5: y-axis units ?

AR: Thank you very much! The y-axis units were lost and now they were reintroduced , and the figure was adjusted.

[Figure]

**Figure 5.** Contributions of different OM sources to the POC measured in the Lena River main stem and the Lena Delta along the transect in 2019.

RC: L592-593: Usually the data are made available prior to submission which is particularly reasonable for open access journals with open review.

AR: We agree with this statement and submitted our data to the PANGEA several months before submitting the manuscript. Unfortunately, due to the current load of PANGEA, our data is still not published yet.

The authors would like to thank AR for their work, time, editing and contribution to our manuscript and wish them all the best!

References used in this response:

Georgiadi, A.G., Tananaev, N.I. & Dukhova, L.A. Hydrochemical Conditions at the Lena River in August 2018. *Oceanology* **59**, 797–800, https://doi.org/10.1134/S0001437019050072, 2019

Geyer, W. R., and D. K. Ralston. 2012. The Dynamics of Strongly Stratified Estuaries. Treatise on Estuarine and Coastal Science. Vol. 2. Elsevier Inc. https://doi.org/10.1016/B978-0-12-374711-2.00206-0

IOCCG (2015). Ocean Colour Remote Sensing in Polar Seas. Babin, M., Arrigo, K., Bélanger, S. and Forget, M-H. (eds.), IOCCG Report Series, No. 16, International Ocean Colour Coordinating Group, Dartmouth, Canada.

Kutscher, L., Mörth, C.-M., Porcelli, D., Hirst, C., Maximov, T. C., Petrov, R. E., and Andersson, P. S.: Spatial variation in concentration and sources of organic carbon in the Lena River, Siberia, J. Geophys. Res. Biogeosci., 122, 1999– 2016, https://doi:10.1002/2017JG003858, 2017

Liu, B., Yang, D., Ye, B., Berezovskaya, S.: Long-term open-water season stream temperature variations and changes over Lena River Basin in Siberia, Global and Planetary

Change, Volume 48, Issues 1–3 (pp 96-111), https://doi.org/10.1016/j.gloplacha.2004.12.007, 2005

Nachimuthu, G., Watkins, M. D., Hulugalle, N., & Finlay, L. A.: Storage and initial processing of water samples for organic carbon analysis in runoff. MethodsX, 7, 101012. https://doi.org/10.1016/j.mex.2020.101012, 2020

Neukermans, G, Ruddick, K., Loisel, H., Roose, P.: Optimization and quality control of suspended particulate matter concentration measurement using turbidity measurements, Limnol. Oceanogr. Methods, 10, doi:10.4319/lom.2012.10.1011, 2012

Röttgers, R., Heymann, K., Krasemann, H.: Suspended matter concentrations in coastal waters: Methodological improvements to quantify individual measurement uncertainty, Estuarine, Coastal and Shelf Science, Volume 151 (pp 148-155), https://doi.org/10.1016/j.ecss.2014.10.010, 2014

Shiklomanov, A.I., R.M. Holmes, J.W. McClelland, S.E. Tank, and R.G.M. Spencer. Arctic Great Rivers Observatory. Discharge Dataset, Version YYYYMMDD. https://www.arcticrivers.org/data, 2021

---

## Editor Decision (ED1)

Editor comments on the revised manuscript BG-2022-183-AT1 "Particulate organic matter in the Lena River and its Delta: From the permafrost catchment to the Arctic Ocean.

I would like to thank the authors for carefully responding to the three reviews and revising the earlier version of their manuscript accordingly.

While cross comparing the Reviewers' comments, the responses, and the corresponding revisions, I still have more requests for revisions. If these required revisions are satisfactory, I could accept it for publication.

L.74: I would like the authors to specify the sampling frequency, rather than simply mentioning "approximately 5–6 samples per year are collected"

L.75: Explain why samples were taken directly from the river's main stems rather than from their deltas and estuaries. The rationale here is not clear.

Overall, I would suggest that Lines 75–87 be rephrased in the logic and details. Here are few examples of suggestions, but I am sure while you consider these, you'll understand how to rephrase it:
- Remove "Thus" at L.75.
- Start a new sentence with "For example…".
- Replace "of the" at L.79 with from the sampling site..
- Replace "and to characterize" with "by characterizing"
- The new addition at L. 76–77 need to be written as an independent sentence, and please add a reference after "one of the world's biggest deltas"

L.98: rephrase as "more than 94% of which is frozen"
L101: remove "and includes". Instead, start with a new sentence "the region is covered by…"
L106: Please use proper reference instead of a link. You could cite this link in the list of references and indicate when it was last accessed.
Also while reading your responses to reviewers, you replied that you defined the boundaries using the hydrosheds classification, and this classification is the most recent. I looked at the suggested ref. in that page and it is 2013, can you explain why you say so? [as Kutscher et al., 2017 is more recent?]
L106: "was defined" instead of "was made"
L120: Please add a reference at the end.
Figure 1:
- I would recommend being consistent with the entire manuscript by using Stolb, rather than Stolb i.
- For the sampling station in the legend and symbols within the map, please remove the red diamond symbol at the sampling station (avoid too many unnecessary colors]. Replace the legend with this as follow (you can fill the markers with black color]:

**Sampling stations:**

▢ main stem

◇ Stolb

⬭ Delta

L131–132: remove "as mentioned above"

L135–137: Please simplify this sentence. It is too wordy

L150: explain why the samples were frozen

L155–156: the word group o the parenthesis is awkward, please rephrase.

L166: How about the river samples? I fact, I did not clearly see any report on how the river samples were analyzed, were they analyzed differently

L177: Typo for HCl

L178: Please add ref. for the tin boats along with dimensions

L178-9: How do you know? I think that a description of this step is skipped here

L182: shouldn't it be reported vs. VPDB?

L184: please add the values for these refs.

L185: and the concentrations were …..

L194: Blank sample was determined…

Also in this paragraph, was pMC–percent modern carbon– included in your analyses? Wouldn't adding it strengthen your data interpretation

L196: add a ref after "for determining OM age"

L197: " $\Delta^{14}C$-depleted" is not grammatically correct. A ratio cannot be depleted

L215: rewrite as Figures 2–5, also Table with T

L223: These are not described in the methods, nor the tools that were used to measure these. Otherwise, please rephrase as "Previous measurements of …. [then add ref]"

L227: remove "In contrast to our surface water samples" and add "instead" after "samples are"

For the newly added text in there, I suggest removing it and keep it for discussion

L233: Since you report this here, this definitely requires you to describe how river water samples were sampled and analyzed in the methods. Please make sure to include that.

L247-8: I agree with this, but you could also emphasize that ArcticGRO database has greater TSM range than yours [if I interpret you figure properly]

Figure 2: It is better to put the ArcticGRO datapoints behind your datapoitss so it is easy to assess where your samples plot vs. ArcticGRO points.

L255: again, reporting river data without method description makes reader doubt about the research, please ensure to add methods pertaining to rivers.

L259: WL19-02 with a value of ….. respectively [also remove the extra "." before the coma

L261: Space forgotten before "The", also please rephrase as " The samples with high TSM…"

L264: why reporting two highest values? Are you referring to a reference threshold?

L270: I believe one of the reviewers commented on this "disembogue". I double checked and it is not a noun. Please use proper noun, e.g., outlet? discharge?]

L280: a value translated to 2236….

L284: Font looks a bit different

L289: rephrase as "values than what we found", then remove "for the ArcticGRO dataset"

L291–2: remove this last sentence, it is redundant

L294: Rephrase as "a strong difference was note don the $\delta^{13}$C of POC. In the Main Stem, $\delta^{13}$C values were…."

L298: The $\delta^{13}$C values of POC…..

L305: The TSM and POC

L317-8: Please indicate ref.

L320: I think "assessed" or "evaluated"? is a better word choice than "analysed"

While I look at Figure 3, the age estimation provided earlier in the manuscript is confusing as there is no more discussion about age I the remaining part of the paper, or I may miss an important information

L.512: "-288 to -122 per mil" if this is so, and while referring to your age estimation above, why not referring to age here?

L525: There was a part similar to this that I suggested to remove earlier, so I suggest keeping this and removing that.

L529: Please avoid as much as possible "etc." when you write scientific paper. Do not let your readers guess. That's a rules of thumb

Figure 4: My question here may be very stupid, but I ask anyway. Do you think that converting ‰ values for the $\Delta^{14}$C is really age appropriate? What is the proportion contribution between Holocene soils and modern OC?

Endmember: This technical term is bothering me. Can you please define what do you really mean by endmembers? In mineralogy, for example, we use endmember as Ca-rich mineral [e.g., anorthite] and Na-rich mineral [e.g., albite], i.e., with a clear geochemical composition, and with a possible predictable mineral with varying Ca an Na composition.

L544. "less depleted in $\delta^{13}$C"—this is an incorrect expression, see similar comment above

L548: Please list these first so it is easy to follow.

L571: this is the reason why I asked earlier if you have pMC data

L581: I don't think that "endmember" is the proper vocabulary here. May be "category"?

L603: replace ";" with "." then start with new sentence "This.."

---

## Author Response (AR2)

**Response to the editor comments on the revised manuscript BG-2022-183-AT1 "Particulate organic matter in the Lena River and its Delta: From the permafrost catchment to the Arctic Ocean."**

Comment types: Authors' Response: "AR", Editor's Comment: "EC"

Comment colors: Authors Response: "blue", Editor's Comment: "black"

Comment fonts: When it was possible, we highlighted changed text by the **bold font**, the text from the manuscript copied to this review was taped *cursive*

AR: Thank you for your review and editing of our manuscript, we highly appreciate your time and work. We have answered all your comments below and revised the manuscript accordingly. There will be a track change version of the manuscript, as well as a clean version including all modifications following your and the 3 anonymous reviewers' suggestions. All the line numbers refer to this clean revised version.

EC: L.74: I would like the authors to specify the sampling frequency, rather than simply mentioning "approximately 5–6 samples per year are collected"

AR: As mentioned in the manuscript this is not our own sampling program, but the Arctic Great Rivers Observatory (Arctic GRO). Unfortunately, we could not find any further information regarding the frequency of sampling rather than to make conclusions on that based on data published by Arctic GRO. This is how we calculated how many samples were taken in each year. We may only suggest that the project aims to capture the seasonal changes, but at the same time sampling is restricted due to the difficulties in accessing the study area. Thus, the Arctic GRO group designed their sampling program such that as many samples as possible are taken in this logistically difficult region while ensuring that enough samples are obtained to describe the seasonal variability of the river. To avoid misreading we changed the sentence:

L70-71: *Within the framework of the ArcticGRO,* ***depending on the year, between 4 and 6*** *samples per year were collected by the ArcticGRO* ***consortium (Holmes et al., 2021)***.

EC: L.75: Explain why samples were taken directly from the river's main stems rather than from their deltas and estuaries. The rationale here is not clear.

AR: Again, since we are not involved in the Arctic GRO project we may only suggest the reason for sampling at Zhigansk. We suggest that this location was chosen due to managing the very

complicated logistics in Arctic regions. A gauging station is situated in Zhigansk city, which likely makes Zhigansk a suitable sampling spot for the ArcticGRO program.

EC: Overall, I would suggest that Lines 75–87 be rephrased in the logic and details. Here are few examples of suggestions, but I am sure while you consider these, you'll understand how to rephrase it:

EC: - Remove "Thus" at L.75.

AR: Changed accordingly

EC: - Start a new sentence with "For example…".

AR: Changed accordingly

EC: - Replace "of the" at L.79 with from the sampling site..

AR: L75: "*This long distance **from** the sampling site **to** the areas where the river enters the Arctic Ocean ....*"

EC: - Replace "and to characterize" with "by characterizing"

AR: Changed accordingly

EC: - The new addition at L. 76–77 need to be written as an independent sentence, and please add a reference after "one of the world's biggest deltas"

AR: Changed accordingly

L74: "*..one of the world's biggest deltas (Fedorova et al., 2015)*"

EC: L.98: rephrase as "more than 94% of which is frozen"

AR: We respectfully disagree with this suggestion because care must be taken and correct terminology must be used when studying permafrost landscapes. 1) 94 % of the Lena River drainage area is not frozen, but it is underlain by permafrost. Every summer the upper part of the soil is thawing forming the active (unfrozen) layer; 2) Extent of permafrost is a crucial parameter for the Arctic region and permafrost zone. Permafrost stores ancient organic matter (OM), which is being released at present due to climate change, resulting in higher OM and nutrient contents in Arctic rivers, modifying food web dynamics and changing the connectivity between terrestrial landscapes and nearshore ecosystems. Permafrost is a key object of this research that is why it is important to mention its extent.

Expressions like "…area/region is underlain by permafrost" are very common in permafrost research, which is used to describe geomorphology in the Arctic regions (for example Kutscher et al., 2017; Strauss et al., 2021; Biskaborn et al., 2019, etc).

Nevertheless, the typo in this sentence was found and edited: L94: "… *94 % is assumed to be underlain **by** permafrost...*"

EC: L101: remove "and includes". Instead, start with a new sentence "the region is covered by…"

AR: We have changed the sentence:

L96-98: *"Running from the south to the north of East Siberia, the Lena River receives OM from various sources within its basin such as Holocene and Pleistocene deposits (Yedoma), which are widespread across the region and cover approximately 3.5 % of the Lena watershed area (Strauss et al., 2013, 2021b)."*

EC: L106: Please use proper reference instead of a link. You could cite this link in the list of references and indicate when it was last accessed.

AR: L107: "…(*Lehner & Grill, 2013*)"

A reference added to the list of references:

*Lehner, B. and Grill, G.: Global river hydrography and network routing: baseline data and new approaches to study the world's large river systems. Hydrol. Process., 27: 2171-2186. https://doi.org/10.1002/hyp.9740, 2013*

EC: Also while reading your responses to reviewers, you replied that you defined the boundaries using the hydrosheds classification, and this classification is the most recent. I looked at the suggested ref. in that page and it is 2013, can you explain why you say so? [as Kutscher et al., 2017 is more recent?]

AR: Given the comments from Reviewer 3 and you, we agree that we should clarify the source and terminology of the subcatchments shown in Figure 1.

The delineations of the subcatchments within the Lena River catchment are based on the HydroSheds database (Lehner et al., 2013). For the purpose of this study, we combined the multiple subbasins following Kutscher et al. (2017) and Liu et al. (2005) to Lower Lena, Aldan, Vilyuy and Upper Lena.

We assume that the data used for the delineation of the subcatchments in Kutscher et al. (2017) is also based on the HydroSheds database, but there might be other sources for catchment areas.

We made changes in our manuscript to clarify the way of delineation of the Lena River subcatchments. In our opinion, the correct way of citing should be:

L100-102: "*Here, we define the Aldan and Vilyuy catchments, the Upper and Lower Lena River by the area of subcatchments of the Lena River using the HydroSheds database (Lehner et al.,*

*2013) and follow the terminology for the subcatchments of Kutscher et al. (2017) and Liu et al. (2005)."*

While there might be publications using a different terminology of the subbasins (as suggested by Anonymous Reviewer 3, including central Lena basin), we were unfortunately unable to find these. Any indication of the source for the disagreement by Reviewer 3 regarding this would have helped.

EC: L106: "was defined" instead of "was made"

AR: Changed accordingly

EC: L120: Please add a reference at the end.

AR: added: L116: *"....(Charkin et al., 2011)"*

EC: Figure 1:

- I would recommend being consistent with the entire manuscript by using Stolb, rather than Stolb i.

AR: We think that it is necessary to mark "Stolb i." in Figure 1 on the map because it is a geographical location: an island named Stolb. Figures 2-4 do not include "i." because they show not a geographical unit but a data category called "Stolb".

To stay consistent we have added " ..Island" for the next lines, if it was missing (where we write about a geographical location): L145, 226, 255, and 465

EC: - For the sampling station in the legend and symbols within the map, please remove the red diamond symbol at the sampling station (avoid too many unnecessary colors]. Replace the legend with this as follow (you can fill the markers with black color]:

**Sampling stations:**

□ main stem

◊ Stolb

○ Delta

AR: We would like to keep the color scheme for filling the symbols of the sampling stations, which we have added following the recommendation from the second anonymous reviewer *("RC2: Figure 1: Include the information on the sample number in the caption. Also, try to showcase three divisions of sample groups for easy understanding").* We have chosen these particular colors to build a connection with other figures throughout the entire manuscript. Data represented on every figure from our paper (except fig.5) are grouped into categories

according to the location of sampling (main stem, Stolb, Delta) and shown as symbols filled with these particular colors.

EC: L131–132: remove "as mentioned above"

AR: Changed accordingly

EC: L135–137: Please simplify this sentence. It is too wordy

AR: Changed accordingly as the sentence was split into two and changed:

L129-132: "*Since the ArcticGRO sampling site is far from where the Lena runoff enters the Arctic Ocean, biogeochemical processes taking place downstream from Zhigansk and particularly in the delta are not reflected in the ArcticGRO data. Thus, the properties of water and suspended materials sampled at Zhigansk may in fact not be entirely representative of the discharge to the ocean.*"

EC: L150: explain why the samples were frozen

AR: L143: Changed accordingly: "…and immediately frozen at -10 °C **for preservation**."

EC: L155–156: the word group o the parenthesis is awkward, please rephrase.

AR: The sentence was changed: L147:"*…We took surface water (0-1 m) at each sampling site.*"

EC: L166: How about the river samples? I fact, I did not clearly see any report on how the river samples were analyzed, were they analyzed differently

AR: The difference in sampling between riverine and deltaic samples is described in detail in chapter 2.2.

L136-142 describe riverine sampling and L142-151 explain the sampling from the delta and in L153-154 information about sampling from previous Lena Delta expeditions is given.

During the laboratory analyses, all samples regardless their origin, were analyzed in the same way. To avoid this misunderstanding, we changed the last sentence from the chapter 2.2:

L154-155: "*This collection of samples and further analyses were conducted in the same way as for the samples from the **riverine** and deltaic transects obtained in 2019.*"

EC: L177: Typo for HCl

AR: Changed accordingly

L178: Please add ref. for the tin boats along with dimensions

AR: L199:"*Then they were dried again for 24 hours at 40 °C and packed/rolled into small tin boats **(6x6x12 mm) (Mollenhauer et al., 2021)**.*"

EC: L178-9: How do you know? I think that a description of this step is skipped here

AR: L170-171: "*For filters with TSM concentrations above 20 mg/L, it was expected that C contents on the filter exceeded 100 µg. For those samples, only a subsample of the filter was used*." "

EC: L182: shouldn't it be reported vs. VPDB?

AR: No, we used Pee Dee Belemnite standard (PBD), but not Vienna PDB standard (VPDB).

EC: L184: please add the values for these refs.

AR: L176: "*National Institute of Standards & Technology, RM 8573, USGS40) with known isotopic composition (–26.39 ± 0.09 ‰).*"

EC: L185: and the concentrations were …..

AR: Changed accordingly

EC: L194: Blank sample was determined…

AR: Changed accordingly

EC: Also in this paragraph, was pMC–percent modern carbon– included in your analyses? Wouldn't adding it strengthen your data interpretation

AR: We report our radiocarbon data in the unit of $\Delta^{14}C$ rather than in related (but uncorrected) units like Fm or pMC. This is common practice and the correct way of referring to data from samples with a known sample collection year. It allows correction for the year of sampling so that data remain directly comparable. We refer to published studies of similar parameters like, e.g., McClelland et al., 2016 in Global Biogeochemial Samples.

Nonetheless, the radiocarbon results were of course first expressed in the unit of fraction modern carbon (F14C or Fm).

EC: L196: add a ref after "for determining OM age"

AR: added: L188:*"…(Stuiver & Polach, 1977)"*

EC: L197: " D14C-depleted" is not grammatically correct. A ratio cannot be depleted

AR: Correct, thanks for noting. For L189 "$\Delta$" was deleted. $\Delta$ and the $\delta$ were also deleted for the same reason in L376, 382, L387:*"…Regardless, 13C- depleted values in POC.."*, L484.

EC: L215: rewrite as Figures 2–5, also Table with T

AR: Changed accordingly

EC: L223: These are not described in the methods, nor the tools that were used to measure these. Otherwise, please rephrase as "Previous measurements of …. [then add ref]"

AR: L212-214: "***Previous measurements using a*** *Conductivity, Temperature and Depth (CTD) probe during the sampling campaign showed no temperature or conductivity stratification of the water profile (Fuchs et al., 2022).*"

EC: L227: remove "In contrast to our surface water samples" and add "instead" after "samples are" For the newly added text in there, I suggest removing it and keep it for discussion

AR: The entire paragraph was removed from the Result chapter. The information from this paragraph was added to the Discussion chapter 4.2.2, L463-466 as suggested.

EC: L233: Since you report this here, this definitely requires you to describe how river water samples were sampled and analyzed in the methods. Please make sure to include that.

AR: Clarification was added to the Methods as it was suggested above.

EC: L247-8: I agree with this, but you could also emphasize that ArcticGRO database has greater TSM range than yours [if I interpret you figure properly]

AR: L233-234:"*TSM reported in the ArcticGRO dataset varied **within a greater range than our main stem samples result** (7.6 and 51.0 mg L$^{-1}$) **nevertheless,** the average TSM (27.8 ±11.3 mg L$^{-1}$) was similar to that of our main stem sample result.*"

EC: Figure 2: It is better to put the ArcticGRO datapoints behind your datapoitss so it is easy to assess where your samples plot vs. ArcticGRO points.

AR: Changed accordantly

[Figure]

EC: L255: again, reporting river data without method description makes reader doubt about the research, please ensure to add methods pertaining to rivers.

AR: Clarification was added to the Methods as it was suggested above.

EC: L259: WL19-02 with a value of ….. respectively [also remove the extra "." before the coma

AR: Changed accordingly

EC: L261: Space forgotten before "The", also please rephrase as " The samples with high TSM…"

AR: Changed accordingly

EC: L264: why reporting two highest values? Are you referring to a reference threshold?

AR: We have changed the sentence: L248: *"The average POC concentration from ArcticGRO is 0.86 ±0.41 mg L-1, **within the range of 0.52 -1.46 mg L-1."***

EC: L270: I believe one of the reviewers commented on this "disembogue". I double checked and it is not a noun. Please use proper noun, e.g., outlet? discharge?]

AR: Edited: L253-254: *"...toward the river mouth (up to 7.1 wt% for LEN19-S-09, sampled at 5 m depth)."*

EC: L280: a value translated to 2236….

AR: changed:

L263-264: *"...Radiocarbon levels of POC varied within a wide range between -243 and -88 ‰ (between 2236 and 740 Δ$^{14}$C years mean age, respectively)"*

EC: L284: Font looks a bit different

AR: Changed accordingly

EC: L289: rephrase as "values than what we found", then remove "for the ArcticGRO dataset"

AR: Changed accordingly

EC: L291–2: remove this last sentence, it is redundant

AR: Changed accordingly

EC: L294: Rephrase as "a strong difference was note don the d13C of POC. In the Main Stem, d13C values were…."

AR: Changed accordingly

EC: L298: The d13C values of POC…..

AR: Changed accordingly

EC: L305: The TSM and POC

AR: Changed accordingly

EC: L317-8: Please indicate ref.

AR: The reference (Shiklomanov et al., 2021b) was added

EC: L320: I think "assessed" or "evaluated"? is a better word choice than "analysed"

AR: changed: L300: *"We **assessed** all ArcticGRO data on TSM and POC for the Lena River to demonstrate that…"*

EC: While I look at Figure 3, the age estimation provided earlier in the manuscript is confusing as there is no more discussion about age I the remaining part of the paper, or I may miss an important information

AR: We provided the age estimation aiming to clarify what we mean when we use such words as "old", "older/younger OM" describing our results within the text. We added this clarification in order to reply to the comment of Anonymous Reviewer 1. The radiocarbon data results were reported as $\Delta^{14}C$ values in ‰, as it described in the Methodology part of the manuscript.

EC: L.512: "-288 to -122 per mil" if this is so, and while referring to your age estimation above, why not referring to age here?

AR: We have chosen to report radiocarbon results as $\Delta^{14}C$ values in ‰, as it is the correct unit allowing comparison between data sets and isotope mass balance calculations. To stay consistent we would prefer not to report these particular parts of our results as age.

EC: L525: There was a part similar to this that I suggested to remove earlier, so I suggest keeping this and removing that.

AR: we have removed the paragraph above and changed the text LL463-466 as:

"*Another explanation for the difference in $\Delta^{14}C$ of POC between ArcticGRO and our riverine transect may be the fact that ArcticGRO samples are depth-integrated, since potentially river water masses may be stratified (e.g. Mackenzie: Hilton et al., 2015). We did not collect samples from different water depths along the river transect from Yakutsk to Stolb but instead were only able to sample surface waters, and from discrete water depths for the samples from the Lena Delta.*"

EC: L529: Please avoid as much as possible "etc." when you write scientific paper. Do not let your readers guess. That's a rules of thumb

AR: Changed accordingly ("etc." was deleted)

EC: Figure 4: My question here may be very stupid, but I ask anyway. Do you think that converting ‰ values for the D14C is really age appropriate? What is the proportion contribution between Holocene soils and modern OC?

AR: All the elements and data on Figure 4 were reported as $\Delta^{14}C$, $\delta^{13}C$ values in ‰, without conversion into age (Endmember values (red crosses) were also given as $\Delta^{14}C$, $\delta^{13}C$ values in ‰ see Table 1). Theoretically, if our manuscript had a different aim it would be possible to convert $\Delta^{14}C$ values into age as it is shown by Stuiver & Polach, 1977.

To answer the second question we would like to refer to Table 2 from our manuscript, where relative OM contribution to POC is represented and/or Figure 5, where this contribution was converted into absolute values of POC concentration (mg L$^{-1}$).

EC: Endmember: This technical term is bothering me. Can you please define what do you really mean by endmembers? In mineralogy, for example, we use endmember as Ca-rich mineral [e.g., anorthite] and Na-rich mineral [e.g., albite], i.e., with a clear geochemical composition, and with a possible predictable mineral with varying Ca an Na composition.

AR: An application of an endmember mixing model for determination of OM sources is a very common method, used and described in multiple publications citied in our manuscript (Mann et al., 2015; Vonk et al., 2010; Wild et al., 2019; Winterfeld et al., 2015).

We provide an explanation how we use the term "endmember" in L206-207: *"Endmember modelling analysis was performed to derive quantitative estimates of the relative input of a potential C source endmember into the POC pool of every water sample and described in detail in 4.2.3."*. Thus, an endmember is a potential C source of POC. In chapter 4.2.3 we discussed how we estimated their isotopic compositions.

EC: L544. "less depleted in d13C"—this is an incorrect expression, see similar comment above

AR: Discussed above and changed accordingly

EC: L548: Please list these first so it is easy to follow.

AR: We have moved this sentence three lines down: L491-492: *"To illustrate possible sources of OM, we used a dual-carbon-isotope (Δ14C, δ13C) three-endmember mixing model. Endmembers for the OM sources in the Lena main stem and its Delta were defined as phytoplankton (I), Holocene soils (II), and Pleistocene deposits (III)…"*

EC: L571: this is the reason why I asked earlier if you have pMC data

AR: Discussed above

EC: L581: I don't think that "endmember" is the proper vocabulary here. May be "category"?

AR: We changed the title of the Table 1: *"Possible sources of C used for the endmember modelling and their isotopic composition (after Winterfeld at al., 2015, Wild et al., 2019, and Galimov at al., 2006)"*

EC: L603: replace ";" with "." then start with new sentence "This.."

AR: Changed accordingly

Reference used:

Biskaborn, B. K., Smith, S. L., Noetzli, J., Matthes, H., Vieira, G., Streletskiy, D. A., Schoeneich, P., Romanovsky, V. E., Lewkow, A. G., Abramov, A., Allard, M., Boike, J., Cable, W. L., Christiansen, H. H., Delaloye, R., Diekmann, B., Drozdov, D., Etzelmüller, B., Grosse, G., Guglielmin, M., Thomas Ingeman-Nielsen, T., Ketil Isaksen, K., Ishikawa, M., Johansson, M., Johannsson, H., Joo, A., Kaverin, D., Kholodov, A., Konstantinov, P., Kröger, T., Lambiel, C., Lanckman, J.-P., Luo, D., Malkova, G., Meiklejohn, I., Moskalenko, N., Oliva, M., Phillips, M., Ramos, M., Sannel, A. B. K., Sergeev, D., Seybold, C., Skryabin, P., Vasiliev, A., Wu, Q.,Yoshikawa, K., Zheleznyak, M., and Lantuit, H.: Permafrost is warming at a global scale, Nat. Commun., 95 10, 264, https://doi.org/10.1038/s41467-018-08240-4, 2019

Mann, P. J., Eglinton, T. I., McIntyre, C. P., Zimov, N., Davydova, A., Vonk, J. E., Holmes, R. M., and Spencer, R. G. M.: Utilization of old permafrost carbon in headwaters of Arctic fluvial networks, Nat. Commun., 6, 7856, https://doi:10.1038/ncomms8856, 2015

Strauss, J., Laboor, S., Schirrmeister, L., Fedorov, A. N., Fortier, D., Froese, D., Fuchs, M., Günther, F., Grigoriev, M., Harden, J., Hugelius, G., Jongejans, L. L., Kanevskiy, M., Kholodov, A., Kunitsky, V., Kraev, G., Lozhkin., A., Rivkina, E., Shur, Y., Siegert, C., Spektor, V., Streletskaya, I., Ulrich, M., Vartanyan, S., Veremeeva, A., Anthony, K.,W., Wetterich, S., Zimov, N., and Grosse, G.: Circum-Arctic Map of the Yedoma Permafrost Domain. Front. Earth Sci. 9:758360, https://doi: 10.3389/feart.2021.758360, 2021

Vonk, J. E., Sánchez-García, L., Semiletov, I., Dudarev, O., Eglinton, T., Andersson, A., and Gustafsson, Ö.: Molecular and radiocarbon constraints on sources and degradation of terrestrial organic carbon along the Kolyma paleoriver transect, East Siberian Sea, Biogeosciences, 7, 3153–3166, https://doi.org/10.5194/bg-7-3153-2010, 2010

Wild, B., Andersson, A., Bröder, L., Vonk, J., Hugelius, G., and Mcclelland, J. W.: Rivers across the Siberian Arctic unearth the patterns of carbon release from thawing permafrost. Proc. Natl. Acad. Sci. U.S.A. 116, 10280–10285, https://doi.org/10.1073/pnas.1811797116, 2019

Winterfeld, M., Laepple, T., and Mollenhauer, G.: Characterization of particulate organic matter in the Lena River delta and adjacent nearshore zone, NE Siberia – Part I: Radiocarbon inventories, Biogeosciences, 12, 3769–3788, https://doi.org/10.5194/bg-12-3769-2015, 2015